

Earth System
Dynamics

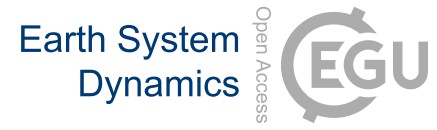

# Atmospheric rivers in CMIP5 climate ensembles downscaled with a high-resolution regional climate model

**Matthias Gröger**[1]**, Christian Dieterich**[1,✝]**, Cyril Dutheil**[1]**, H. E. Markus Meier**[1,2]**, and Dmitry V. Sein**[3,4]

[1]Department of Physical Oceanography and Instrumentation, Leibniz Institute for Baltic Sea Research
Warnemünde, Rostock, Germany
[2]Research and Development Department, Swedish Meteorological and Hydrological Institute,
Norrköping, CE1 Sweden
[3]Shirshov Institute of Oceanology, Russian Academy of Sciences; Moscow, Russia
[4]Alfred Wegener Institute, Helmholtz Centre for Polar and Marine Research, Bremerhaven, Germany
✝deceased, TS2

**Correspondence:** Matthias Gröger (matthias.groeger@io-warnemuende.de)

**Abstract.** TS3 Atmospheric rivers (ARs) are important drivers of hazardous precipitation levels and are often associated with intense floods. So far, the response of ARs to climate change in Europe has been investigated using global climate models within the CMIP5 framework. However, the spatial resolution of those models (1–3°) is too coarse for an adequate assessment of local to regional precipitation patterns. Using a regional climate model with 0.22° resolution, we downscaled an ensemble consisting of 1 ERA-Interim (ERAI) reanalysis data hindcast simulation, 9 global historical, and 24 climate scenario simulations following greenhouse gas emission scenarios RCP2.6, RCP4.5, and RCP8.5.

The performance of the climate model to simulate AR frequencies and AR-induced precipitation was tested against ERAI. Overall, we find a good agreement between the downscaled CMIP5 historical simulations and ERAI. However, the downscaled simulations better represented small-scale spatial characteristics. This was most evident over the terrain of the Iberian Peninsula, where the AR-induced precipitation pattern clearly reflected prominent east–west topographical elements, resulting in zonal bands of high and low AR impact. Over central Europe, the models simulated a smaller propagation distance of ARs toward eastern Europe than obtained using the ERAI data.

Our models showed that ARs in a future warmer climate will be more frequent and more intense, especially in the higher-emission scenarios (RCP4.5, RCP8.5). However, assuming low emissions (RCP2.6), the related changes can be mostly mitigated. According to the high-emission scenario RCP8.5, AR-induced precipitation will increase by 20 %–40 % in western central Europe, whereas mean precipitation rates increase by a maximum of only 12 %. Over the Iberian Peninsula, AR-induced precipitation will slightly decrease ($\sim$ 6 %) but the decrease in the mean rate will be larger ($\sim$ 15 %). These changes will lead to an overall increased fractional contribution of ARs to heavy precipitation, with the greatest impact over the Iberian Peninsula (15 %–30 %) and western France ($\sim$ 15 %). Likewise, the fractional share of yearly maximum precipitation attributable to ARs will increase over the Iberian Peninsula, the UK, and western France.

Over Norway, average AR precipitation rates will decline by −5 % to −30 %, most likely due to dynamic changes, with ARs originating from latitudes > 60° N decreasing by up to 20 % and those originating south of 45° N increasing. This suggests that ARs over Norway will follow longer routes over the continent, such that additional moisture uptake will be impeded. By contrast, ARs from > 60° N will take up moisture from the North

Please note the remarks at the end of the manuscript.

Atlantic before making landfall over Norway. The found changes in the local AR pathway are probably driven by larger-scale circulation changes such as a change in dominating weather regimes and/or changes in the winter storm track over the North Atlantic.

## 1   Introduction

Atmospheric rivers (ARs) are long, narrow corridors that transport enormous amounts of moisture from tropical and subtropical origins poleward (e.g., Zhu and Newell, 1998 TS4; Gimeno et al., 2014, 2016; Shields et al., 2019). Due to their intense moisture loads, they play an important role in the global water cycle. It has been estimated that ARs are responsible for $> 90\%$ of meridional moisture transport through midlatitudes (e.g., Gimeno et al., 2016, 2018 TS5). In addition, ARs are associated with very powerful low-level winds often positioned at the head of a cold front of an extratropical storm system (e.g., Dacre et al., 2015; Gimeno et al., 2016). Accordingly, they are modulated by large-scale weather regimes, as demonstrated by Pasquier et al. (2019). In the North Atlantic sector, the moisture contained in ARs originates mainly from the subtropical Atlantic (Ramos et al., 2016a). Although ARs can occur throughout the year, due to their strong linkage to extratropical storm systems they are more frequent during the cold season in the Northern Hemisphere (Lavers and Villarini, 2013; Ramos et al., 2015).

In the North Atlantic and North Pacific, ARs pose a serious risk of heavy precipitation and flooding along the western coasts of California and Europe (e.g., Ralph et al., 2006; Neiman et al., 2011; Ralph and Dettinger, 2012; Lavers et al., 2011, 2012; Lavers and Villarini, 2013; Ramos et al., 2015; Gao et al., 2016; Nayak et al., 2016; Nayak and Villarini, 2017 TS6). Flooding is expected to increase under a warming climate, incurring high economic costs as well (Ashley et al., 2005; Sayers et al., 2015; Alfieri et al., 2017 TS7). Elucidation of the mechanisms that give rise to ARs and thus to an increased flood risk is therefore essential to mitigate their impact (e.g., Kousky, 2014; Alfieri et al., 2018).

Several studies have examined heavy-precipitation events and flooding in Europe attributable to ARs. For example, Lavers and Villarini (2013) analyzed atmospheric reanalysis data and found that between 1979 and 2011 as many as 8 of the 10 annual maximum precipitation events were related to ARs. In Europe, most damage associated with ARs occurs along the western continental margin, especially over the Iberian Peninsula, the UK, and Scandinavia (Lavers et al., 2013; Ramos et al., 2015; Whan et al., 2020). However, ARs can also penetrate far inland, producing heavy rainfall events as far east as Germany and Poland (Lavers and Villarini, 2013; Ionita et al., 2020). Unlike the heavy-precipitation events associated with the local formation of short-lived convective cells during summer, ARs typically produce hazardous precipitation continuously over several days (Shields

and Kiehl, 2016 TS8). Nonetheless, with their intense precipitation, ARs also strongly contribute to local groundwater management. In dry and semi-arid regions, they can play an important role in local water groundwater recharge and the irrigation of dry land vegetation (Albano et al., 2017) as is the case in many regions around the Mediterranean (Martos-Rosillo et al., 2015).

Because of the larger water-holding capacity characteristic of a warmer atmosphere, climate warming is expected to increase the risk of intense flooding (e.g., Held and Soden, 2006). Lavers et al. (2015) demonstrated that an intensification of the global water cycle due to climate warming will strengthen the mean transport rate of atmospheric water over the North Atlantic by 30 %–40 %. So far, assessments of ARs in a future warmer climate have been primarily based on climate projections from global models (e.g., Lavers et al., 2013; Warner et al., 2015; Ramos et al., 2016b; Gao et al., 2016; Espinoza et al., 2018; Whan et al., 2020). For example, Lavers et al. (2013) analyzed five global models from the Couple Model Intercomparison Project (CMIP5, Taylor et al., 2012) and found an intensification of ARs in terms of their frequency and moisture load in a future climate. Based on the RCP4.5 and RCP8.5 scenarios, Ramos et al. (2016b) determined a doubling of AR frequency together with an increased moisture load at the end of the 21st century compared to the historical period. Gao et al. (2016) analyzed an ensemble of 24 CMIP5 global models and found a pronounced increase in the fractional contribution of AR-induced precipitation to the total annual precipitation, based on global projections following the RCP8.5 scenario. Whan et al. (2020) used the high-resolution version of the CMIP5 EC-Earth model to study the impact of climate change on AR-induced precipitation over Norway. Up to 80 % of the winter maximum precipitation was shown to be associated with ARs. The authors also found that the magnitude of extreme precipitation events is mainly controlled by AR intensity.

The aforementioned studies analyzed global models from the CMIP5 and CMIP6 frameworks, in which the spatial resolution typically ranges from 1 to 3°. This resolution is sufficient to assess the large-scale impact of climate on precipitation, but it is unable to fully resolve small-scale characteristics such as small convective cells (Hoheneger et al., 2020; Stevens et al., 2020). A further shortcoming of global models is their poor representation of orography, which in both CMIP5 and CMIP6 is typically lower than in the real world and thus leads to the oversimplifications of modeled processes (Baldwin et al., 2021) associated with, e.g., the uplift or blockage of an air mass. While high-resolution climate

models covering a limited region largely settle these issues, regional model assessments that focus on ARs are still lacking for Europe. Therefore, in this study, a high-resolution regional climate model for Europe was employed to downscale global climate simulations derived from the CMIP5 suite.

The main purposes of this study are to

- conduct the first analysis of ARs over Europe using a downscaled CMIP5 model ensemble,

- investigate the added value of high resolution in representing ARs in a climate model,

- assess future climate-related changes in AR characteristics over Europe, and

- explore uncertainties with respect to the choice of the global model and in regard to the choice of the greenhouse gas (GHG) emission scenario.

In the following, we present the first analysis of ARs in a regional climate ensemble for Europe based on a horizontal resolution of 0.22°. The ensemble was used to examine future changes in AR frequency and AR-induced heavy-precipitation patterns over Europe as well as the impact of ARs on the local water budget. Climate-induced changes in the pathways of ARs on their journey across Europe were analyzed. Finally, uncertainties with respect to three different climate scenarios (RCP2.6, RCP4.5, RCP8.5) and nine different parent global climate models from the CMIP5 suite were assessed.

The paper is structured as follows: Sect. 2 briefly presents the regional climate model (RCA) and the AR detection procedure. Section 3 describes the validation of the climate model, based on a comparison of downscaled CMIP5 historical simulations with ERA-Interim (ERAI) reanalysis data. The added value of a high resolution is demonstrated by downscaling the ERAI reanalysis (Dee et al., 2011) from 0.75° resolution to 0.22° using the high-resolution model RCA. Section 4 analyzes future changes in AR frequencies and the impact on precipitation under different climate scenarios. Section 5 discusses uncertainties with respect to the choice of the driving global model. Our main conclusions make up Sect. 6.

## 2 Methods

### 2.1 The regional climate model RCA

The atmospheric part of the regional climate model (Wang et al., 2015; Gröger et al., 2015; Dieterich et al., 2019 `TS9`) is based on Rossby Center regional atmosphere (RCA) model (Samuelsson et al., 2011; Kupiainen et al., 2014) version 4. RCA was set up for the European Coordinated Regional Climate Downscaling Experiment (EURO-CORDEX) domain (Fig. 1). The horizontal resolution is 0.22° on a rotated grid, which results in a metric resolution of ∼ 24 km (Table 1).

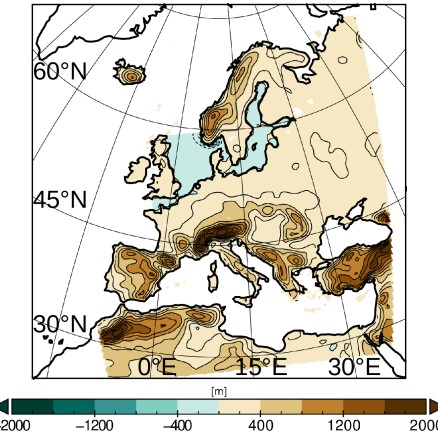

**Figure 1.** Model domain showing the land topography of the RCA climate model (in shades of brown). Bathymetry information is within the domain of the interactively coupled ocean model NEMO (in shades of blue).

The vertical resolution is given by 40 hybrid levels. At the lateral boundaries, the model is driven either by reanalysis data (ERAI, Dee et al., 2011) or global climate model output. Hence, there is no feedback from the RCA to outside the model domain. This means that ARs enter the model domain from the parent global model but then develop freely and independently of the model. The forcing data are prescribed at 6-hourly time intervals.

The land–surface boundary is defined according to ECO-CLIMAP (Champeaux et al., 2005) and used to calculate mass and energy fluxes between the Earth's surface and the atmosphere. Over the North Sea and Baltic Sea, RCA is interactively coupled to the 3-D ocean model NEMO (Nucleus for European Modelling of the Ocean; Madec, 2012, Fig. 1, Table 1). Sea ice temperature, sea ice fraction, sea ice albedo, and water temperature over this region are explicitly modeled by NEMO. Air–sea mass and energy fluxes are then calculated in the atmosphere model and used to drive NEMO, which is set up at a resolution of 2 nautical miles (∼ 3.7 km) and 56 vertical varying $z^*$ layers. Coupling is managed by the OASIS coupler (Valcke et al., 2003). However, with its high resolution and short time stepping, NEMO is very expensive to run. Therefore, outside the coupled domain, i.e., the Mediterranean and North Atlantic, RCA is driven by reanalysis data or global climate model output.

The climate model RCA has been intensively validated and comprehensively described (e.g., Wang et al., 2015; Gröger et al., 2015, 2019, 2021a; Dieterich et al., 2019 `TS10`). It has been employed in previous studies to investigate the present climate and simulate the mean response to global climate change by downscaling global climate scenarios (Dieterich et al., 2019 `TS11`; Gröger et al., 2019, 2021a). Gröger et al. (2021a) showed that the RCA-NEMO coupled ensemble is well within the range of the high-resolution EURO-CORDEX ensemble (Jacob et al., 2014). However, signifi-

**Table 1.** The climate model RCA configuration employed in this study. The analyzed data set from the ERAI reanalysis is also shown (Dee et al., 2011).

| Model system | Atmosphere component | Ocean component | Domain size atmosphere | Domain size ocean model | Grid resolution atmosphere | Vertical levels | Temporal resolution |
|---|---|---|---|---|---|---|---|
| RCA | RCA | NEMO3.3.1 | EURO-CORDEX | North Sea, Baltic Sea | $0.22°$ ($\sim 24 \times 24$ km) | 40 | 6 h |
| ERAI | IFS (cy31r2) | Prescribed SSTs | Global | Global | $0.75°$ ($80 \times 45$ km) | 60 | 6 h |

**Table 2.** Overview of the 34 regional RCA simulations grouped by GHG emission scenarios and downscaled global models. For validation purposes, an RCA-ERAI hindcast was carried out for subsequent comparison with the ERAI hindcast data set. Due to technical issues, RCP2.6 scenarios for RCA – IPSL-CM5A-MR, RCA – CanESM2, and RCA – CNRM-CM5 could not be performed.

| Reg. model – glob. model | Historical (1970–1999) | RCP2.6 (1970–1999) | RCP4.5 (1970–1999) | RCP8.5 (1970–1999) |
|---|---|---|---|---|
| RCA – CANESM2 | x | | x | x |
| RCA – CNRM-CM5 | x | | x | x |
| RCA – EC-Earth | x | x | x | x |
| RCA – GFDL-ESM2M | x | x | x | x |
| RCA – HadGEM2-ES | x | x | x | x |
| RCA – IPSL-CM5A-MR | x | | x | x |
| RCA – MIROC5 | x | x | x | x |
| RCA – MPI-ESM-LR | x | x | x | x |
| RCA – NorESM1-M | x | x | x | x |
| Hindcast period | (1979–2008) | | | |
| RCA – ERAI | x | | | |
| ERAI – (ECMWF-IFS) | x | | | |

cant differences arise for interactively coupled areas over the sea (Gröger et al., 2021a, b). This applies to both climatic mean changes and climatic extremes (e.g., dry periods, cold spells, heat waves).

## 2.2 The high-resolution climate ensemble

The above-described model was used to downscale a set of global model climate scenarios taken from the CMIP5 suite. Table 2 lists the downscaled RCA simulations as well as the applied scenarios (first row) and individual model configurations (first column).

The chosen climate scenarios follow the protocol of the Representative Concentration Pathways (RCPs) used in CMIP5 and derive from three different GHG emission assumptions. The low-emission scenario RCP2.6 assumes both vigorous mitigation actions (van Vuuren et al., 2007, 2011) to limit the global mean temperature increase to $+2\,°C$ compared to the pre-industrial period and negative emissions during the last decade of the 21st century. RCP4.5 is a moderate-emission scenario in which emissions peak at mid-century (2040) and remain constant after $\sim 2080$, at a value about half of that at the end of the historical period (Clarke et al., 2007; Thomson et al., 2011 TS12). In the totally unmitigated scenario (RCP8.5) (Riahi et al., 2007, 2011), rising emissions until the end of the century are assumed. The three scenarios impose a maximum radiative forcing of 2.6, 4.5, and $8.5\,W\,m^{-2}$ compared to pre-industrial conditions.

In addition to the climate simulation, we produced a hindcast run with RCA forced by ERAI reanalysis data at the lateral boundaries (RCA-ERAI, Dee et al., 2011, Table 2). In this study, the hindcast was compared with the original ERAI data, which have a resolution of only $0.75°$ but the same temporal resolution of 6 h (Table 1). The original ERAI data were interpolated onto the grid of the RCA climate model using the bilinear remapping technique provided by the Climate Data Operators software package (Schulzweida, 2021). This comparison demonstrated the added value of downscaling. For a full description of the ERAI reanalysis data set, the reader is referred to Dee et al. (2011).

## 2.3 Detection of atmospheric rivers

A number of studies have addressed methods to detect ARs, based on model simulations in a Eulerian framework (Lavers et al., 2011, 2012; Nayak et al., 2014; Nayak and Villarini, 2017 TS13; Gao et al., 2015; O'Brien et al., 2020). An overview of some of those methods can be found in Shields et al. (2018). In our study, we employed the detection algorithm developed by Lavers et al. (2012) and Lavers and Villarini (2013), as it has been successfully applied in hindcast simulations and in future projections (Lavers and Villarini,

2013 TS14; Lavers et al., 2013). In this algorithm, vertically integrated atmospheric water vapor transport ([kg m$^{-1}$ s$^{-1}$], hereafter IVT) is calculated at every 6-hourly model output time step. The vertical integration is done over pressure levels ranging from 1000 to 300 hPa (Lavers et al., 2012):

$$\text{IVT} = \sqrt{\left(\frac{1}{g}\int_{1000}^{300} qu\,dp\right)^2 + \left(\frac{1}{g}\int_{1000}^{300} qv\,dp\right)^2}, \quad (1)$$

where $g$ is gravitational acceleration [m s$^{-2}$], $q$ is specific humidity [kg kg$^{-1}$], $u$ and $v$ are horizontal wind components [m s$^{-1}$], and $dp$ [Pa] is the pressure level difference of adjacent pressure levels.

Next, the detection algorithm is launched (Lavers and Villarini, 2013). It is composed of the following steps:

1. From the 6-hourly IVT time series between 1970–1999, all time steps at 12:00 UTC are extracted.

2. Along 10° W, all IVT is sampled separately in seven meridional 5° bins between 35–70° N, i.e., 35–40, 40–45, ... 65–70° N. Hence, the sample size for each meridional 5° bin consists of $\sim 30$ (years) $\times 365$ (12:00 UTC) $\times N$ (number of grid cells in the respective 5° bin at 10° W).

3. For each of the seven bins, the 85th percentile IVT is calculated. The 85th percentile serves then as the threshold in the detection of ARs (Lavers and Villarini, 2013 TS15; Fig. 2).

After the bin-specific IVT thresholds were determined, the entire IVT time series containing all time steps (00:00, 06:00, 12:00, 18:00 UTC TS16) is searched for ARs.

4. ARs are detected separately for each of the seven 5° latitudinal bins along 10° W and at every time step. If the max IVT within the respective bin exceeded the threshold for that bin (Fig. 2) a search is conducted from 10° W westward to 30° W and eastward to 25° E. All grid cells in which the threshold is exceeded are retained together with its time stamps (Lavers and Villarini, 2013). We note that the detection using the 85th percentile at 10° W is well validated for ARs over the Atlantic. However, it may lead to some inaccuracies over the eastern European land mass. This is because the AR landfall at the western European boundary may decrease the IVT along the eastern landmass due to moisture loss by precipitation and the moisture cutoff from the ocean. This could limit the detection of AR impact over the distant parts in eastern Europe. A potential solution could be to take the local 85th percentile over land points instead the 85th percentile at 10° W as threshold. However, this should be robustly tested and validated in future research.

5. The resulting AR time series is then further evaluated according to spatial and temporal criteria (Lavers and Villarini, 2013). Hence, the axis of a potential ARs is determined as a maximum IVT value along subsequent longitudes and the total length is calculated. Following Lavers and Villarini (2013) and Lavers et al. (2015), in our study only those fields in which the AR axis was longer than 1500 km were retained. Due to our limited domain, the algorithm does not detect ARs that do not reach Europe but remain out over the Atlantic Ocean. Thus, across the western Iberian Peninsula, which is located relatively close to the model's western boundary, some ARs might have been missed or detected with a delay (as it may take longer to reach the 1500 km criterion when the AR proceeds into the model). Over the UK and Norway, this did not have a significant effect as these countries lie far away from the model's lateral boundary.

6. At this stage, the retained AR fields can contain more than one AR at every time step because an AR may cover two adjacent bins. These double entries are removed.

7. Finally, ARs are checked for "lifetime". ARs must have a lifetime of at least 18 h, corresponding to three or more consecutive 6-hourly output time steps. All other time steps were discarded.

8. During the detection, for the purpose of post-analysis, AR masks are generated and archived at a resolution of 6-hourly time steps during the lifetime of the AR. The masks are based on the exceedance of the bin thresholds along 10° W (as shown in Fig. 2) and contain information on the moisture content as well as the date and time (Fig. 3). This allows calculation of the mean IVT within an AR. The masks are also used to calculate the mean precipitation associated with ARs and to analyze the routes taken by the ARs over the European continent. TS17

Note that for the ERAI data set and the RCA-ERAI (Table 2) hindcast simulation, the analysis period is 1979–2008, as ERAI data from before 1979 are not available.

Figure 3 provides an example of an AR that caused intense rain over France and Germany and was detected in the ERAI reanalysis (left) and in the ERAI hindcast simulation (right). The detection procedure is performed separately for the historical and future periods and for each model (Table 2).

## 2.4 Detection in future climate

Unlike previous studies (e.g., Lavers et al., 2013; Gao et al., 2016; Ramos et al., 2016), in this study we did not use the historical thresholds to detect ARs in a future climate. Instead, the IVT thresholds were calculated based on the respective climatologies between 2070 and 2099. This was

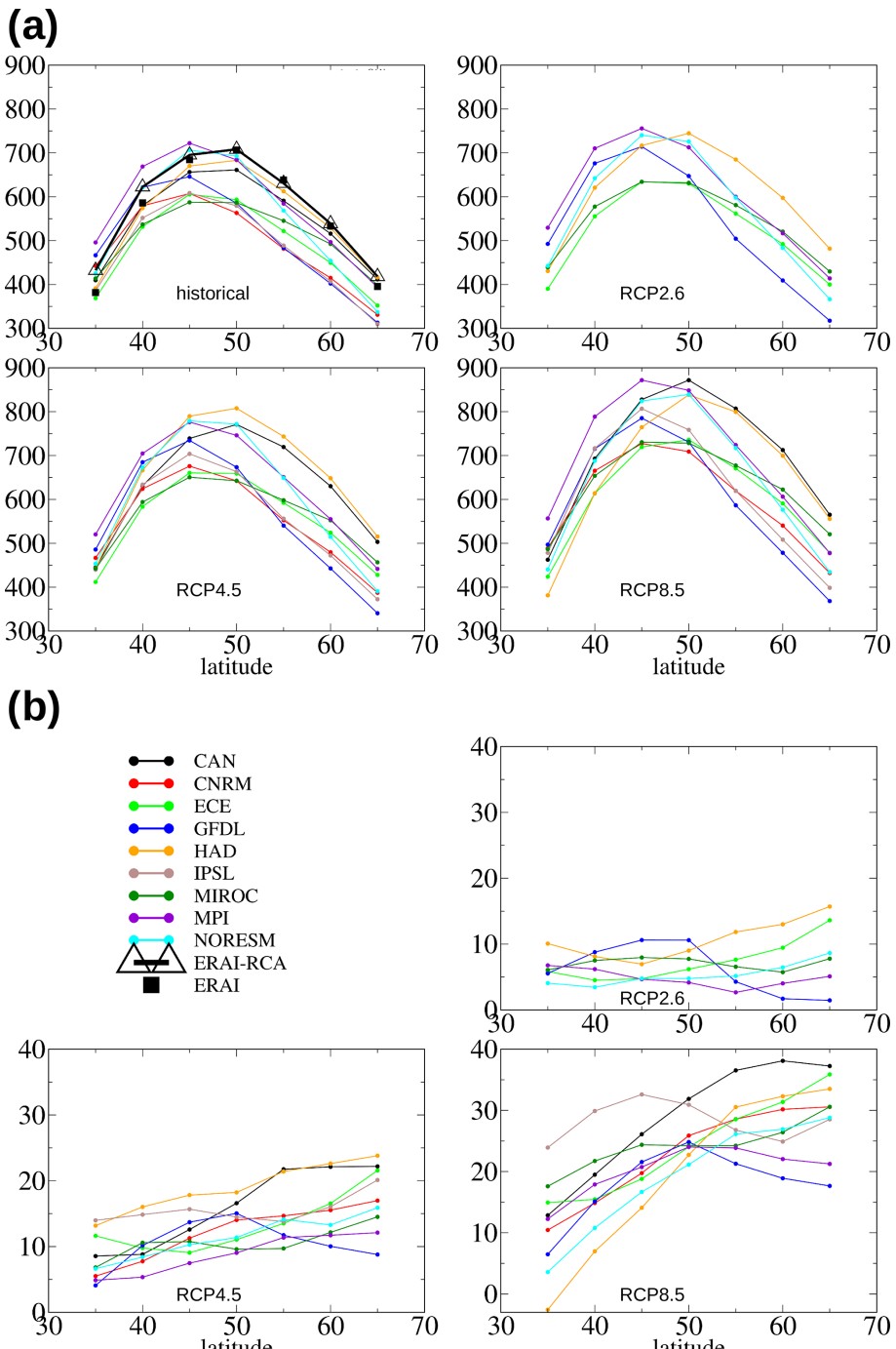

**Figure 2. (a)** The 85th percentiles of IVT [kg m$^{-1}$ s$^{-1}$] at 10° W for all models and the historical period (1970–1999, upper left) and the RCP climate scenarios (2070–2099). The algorithm uses the values to find ARs. **(b)** Relative change [%] in the IVT thresholds in future scenarios relative to the respective historical period.

done to keep the empirical relationship between the 85th percentile IVT and the moisture content in ARs derived for the present climate. Hence, according to weather data for the years 1998–2005, the 85th percentile at noon (12:00 UTC ₅ TS18 ) corresponded roughly to the median moisture content

of the observed ARs (see Lavers and Villarini, 2013, for details).

Figure 2 shows that the 85th percentile can strongly increase in the future climate depending on the scenario, the latitude, and the respective model. An example is RCA- ₁₀ CAN for which the IVT threshold increases by nearly 40 %

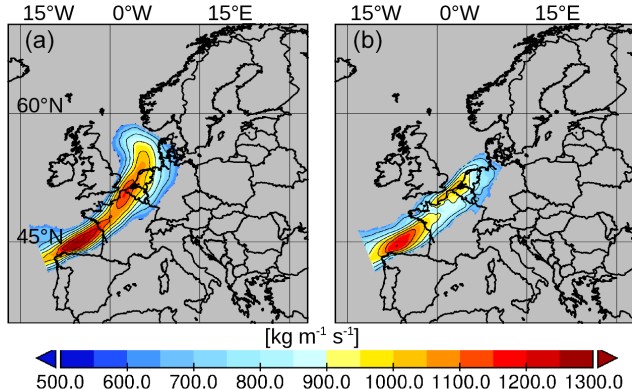

**Figure 3.** AR associated with storm Lothar, detected in the ERAI reanalysis (**a**) and in the ERAI hindcast simulation (**b**), 25 December 1999; 00:00:00 UTC. IVT values below the AR 85th percentile are masked out.

in RCP8.5 north of 55° N (Fig. 2b). In this simulation, the 85th percentile IVT from the historical run at 55° N corresponds to the 68th percentile in the future period.

Hence, in this study, a stable empirical relationship between the 85th percentile and the median was assumed. Consequently, our approach limits the influence of the larger mean atmospheric moisture content in the future climate and instead emphasizes dynamic changes.

## 2.5   Calculation of indices

The impact of ARs under present and future climate was investigated by calculating the following indices:

1. ARF is the AR frequency expressed as a percent of AR days per total days within a 30-year period. A calendar day is classified as an AR day if within the 24 h at least one AR event is recognized. Even if only one of the four 6-hourly time steps during the day is impacted by an AR, the day is counted as an AR day. Consequently, an AR lasting 18 h and extending over 2 d is counted as 2 AR days. No distinction was made regarding whether an AR day was associated by precipitation or not.

2. % AMP is the percent annual maximum precipitation rate related to ARs. For every land point, the maximum precipitation rate for every year is extracted together with the date and time of its occurrence. Then every grid cell is checked to determine whether this annual maximum is related to an AR at this time and position (using the aforementioned AR masks). Hence, if 15 out of the 30 annual maxima are attributable to ARs, the % AMP is equal to 50 %.

3. % 95P is the fractional contribution (%) of AR-related precipitation to the total heavy precipitation (precipitation events > 95th percentile precipitation).

i. The precipitation is summed if the associated precipitation rate exceeds the 95th percentile precipitation calculated from all precipitation events.

ii. This is repeated but only for those precipitation rates related to ARs. Thus, the % 95 P is the percent contribution of AR-related precipitation to the total heavy precipitation.

4. % TP is the same as the % 95P, but all precipitation events are considered. Thus, the % TP is the percent contribution of AR-related heavy precipitation to the total precipitation.

## 3   ARs in the historical simulations and hindcasts

### 3.1   Differences in IVT thresholds in historical and hindcast simulations

A comparison of the latitude-dependent IVT thresholds of the RCA simulations with those derived from ERAI is shown in Fig. 2a. As expected, the RCA-ERAI (triangles in Fig. 2a) hindcast simulation was closest to the ERAI reanalysis (filled squares). Notable discrepancies between the two data sets occurred at latitudes 40 and 35° N. Due to the rotated grid of the RCA, the positions of 35 and 40° N at 10° W were closest to the western lateral boundary of the RCA (Fig. 1). The discrepancies most likely stem from known issues with lateral boundary coupling, as occurs in limited area models with one-way coupling to the global models (e.g., Davies, 2014 TS19; Chikhar and Gauthier, 2017). Moreover, the southern model boundary at 35° N lies in the transition zone between dry air masses from the subtropics and the wet air masses of the westerlies. In this transition zone, large gradients in moisture content over short distances can be expected. Consequently, small differences in the mean position of the transition zone can cause large differences in the local moisture content.

The pronounced differences that characterize the different RCA historical simulations (Fig. 2a) reflect the different climates derived from the parent global models at the lateral boundary. First, the parent global models differ in their thermodynamic equilibrium states, such that both the air temperatures and the moisture loads at the lateral boundary of the RCA differ as well. As a result, the large-scale atmospheric circulation also differs among the global models (e.g., Brands, 2022), as the Equator-to-pole temperature gradients are likewise different (Harvey et al., 2014). This further influences the meridional position of the westerlies in the respective models. For example, the maximum moisture transports are located around 50° N in RCA-HAD, RCA-CAN, and RCA-MIROC but at 45° N in the other historical runs.

There was no evidence of a linear relationship between climate warming and the increase in IVT thresholds (Fig. 2b). For example, in RCA-HAD the IVT at 35° N in the low

**Table 3.** Number of ARs detected in a 30-year period of different climate scenarios. The historical period for all runs is 1970–1999. The exception is the ERAI run, for which it was 1979–2008. Numbers in parentheses denote the relative change (%) in the future vs. the historical period.

| | Historical/ hindcast | RCP2.6 | RCP4.5 | RCP8.5 |
|---|---|---|---|---|
| RCA-ERAI | 322 | | | |
| ERAI reanalysis | 321 | | | |
| RCA climate ensemble | | | | |
| RCA-MEAN | 359 | 390.0 (+8.6) | 425.7 (+18.6) | 445.6 (+24.1) |
| RCA-STD | 58 | 67.4 (+16.46) | 71.50 (+23.6) | 90.3 (+56.0) |
| RCA-CAN | 367 | | 445 (+21.3) | 447 (+21.8) |
| RCA-CNRM | 285 | | 317 (+11.2) | 362 (+27.0) |
| RCA-ECE | 393 | 396 (+0.76) | 412 (+4.8) | 468 (+19.1) |
| RCA-GFDL | 351 | 367 (+4.6) | 388 (+10.5) | 456 (+29.9) |
| RCA-HAD | 397 | 422 (+11.3) | 484 (+21.9) | 565 (+42.3) |
| RCA-IPSL | 409 | | 498 (+21.8) | 523 (+27.9) |
| RCA-MPI | 304 | 311 (+2.3) | 350 (+15.1) | 392 (+29.0) |
| RCA-MIROC | 262 | 264 (+0.76) | 302 (+15.3) | 276 (+5.3) |
| RCA-NorESM | 421 | 417 (−0.91) | 457 (+8.6) | 521 (+23.8) |

and moderate warming scenarios (RCP2.6 and RCP4.5) increased by ∼ 10 % and ∼ 15 %, respectively (Fig. 2b). In the strongest warming scenario (RCP8.5), however, the IVT decreased by ∼ 5 %. This suggests that, over the long term, dynamic changes influence the IVT. Also decadal variations in wind components and specific humidity have to be considered in this context (Thandlam et al., 2022 TS20).

## 3.2 General statistics

Table 3 summarizes the number of ARs for each run of the climate ensemble and the ERAI reanalysis data set and for the RCA-ERAI hindcast run. The results were almost identical for the RCA-ERAI hindcast (322) and ERAI reanalysis itself (321), which was used to drive the RCA. This indicates that the number of ARs in the RCA is primarily controlled by the parent global model at the lateral boundaries. This is not surprising since ARs develop in open-ocean regions far outside the model's domain. However, within that domain, the RCA develops freely, leaving its own fingerprint on ARs, by controlling their intensity, geometry, and lifetime (Fig. 3). The RCA fingerprint is likely to become stronger with growing distance from the lateral boundaries. Analogously, ARs in the respective RCA climate simulations (Table 3) will reflect the ARs generated from the driving global climate model. Consequently, the RCA historical climate ensemble had a fairly large spread during the historical period, ranging from 262 (RCA-MIROC) to 421 (RCA-NorESM). The difference between the ensemble mean (RCA-MEAN) of the historical simulations (RCA-MEAN = 359) and the RCA-ERAI hindcast run ($n = 322$) was small compared to the standard deviation over the RCA historical ensemble (58).

Compared to the historical period, RCA-MIROC increased by 15 % in RCP4.5 but only by 5 % in the strongest warming scenario (RCP8.5, Table 3) which is lowest increase in the ensemble. Hence, there was no linear scaling with global mean warming compared to the other models. This can be explained by the changes in the large-scale circulation in the parent global model, such as induced by shifts in the eddy-driven jet (Gao et al., 2016).

A comparison of the moisture transported by ARs over land is shown in Fig. 4a, which depicts the potential of ARs to force local heavy-precipitation events. Note that the moisture content over land is lower in the RCA-ERAI run than in the ERAI (∼ 5 %, Fig. 4a). This is in line with the model's cold bias in air temperature (Gröger et al., 2021a), thus favoring a lower moisture content. The lower moisture content in RCA-MEAN than in the RCA-ERAI hindcast simulation should also be pointed out. Overall, this suggests a systematic negative bias in the moisture content over land in the RCA model. The distribution of diagnosed AR durations (Fig. 4b) does not indicate systematic differences between the ERAI reanalysis, the hindcast run, and the mean historical climate simulations. For all model realizations, about half of the detected ARs lasted 1 d or less (Fig. 4b).

## 3.3 Impact on precipitation

Maps of ARFs over land are presented in Fig. 5a. As expected, during the historical period, ARs were most abundant over the UK and the coastal regions of western Europe. Further inland, AR frequencies declined as the ARs lost moisture due to rainfall and thus no longer met the IVT threshold. Strong moisture losses also occurred along the Norwegian

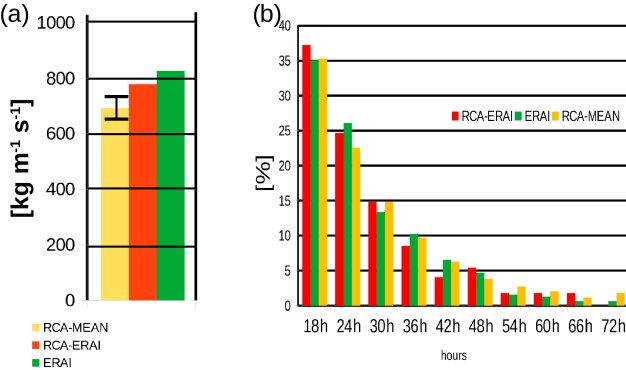

**Figure 4. (a)** Average AR moisture transport over land for each model realization depicted from the climatological historical period (RCA-MEAN) as well as for the hindcast simulation (RCA-ERAI) and the reanalysis data set (ERAI). For RCA-MEAN, the range of 2 standard deviations from individual model simulation is given. **(b)** Histogram of average durations of detected ARs.

coast, where AR landfalls caused heavy rain events due to orographic uplift.

We then evaluated the potential of ARs to cause annual maxima of daily precipitation (Fig. 5b, % AMP). Locally, ARs explained up to ∼ 60 % of the yearly maxima over southwestern Norway. A strong imprint was likewise seen over the western UK and along European coasts, where ARs were responsible for up to 50 % of the annual maxima in RCA-MEAN and RCA-ERAI.

Besides their potential to cause yearly precipitation maxima, ARs give rise to heavy-precipitation events. The fractional contributions of ARs to heavy precipitation (% 95P) and to the total precipitation (% TP) are shown in Fig. 5c and d. The spatial pattern mainly mirrored the AR frequency pattern, but it also reflected the varying long-term mean hydrological conditions in Europe: in semi-arid regions such as the Iberian Peninsula and along the western coast of Italy, % 95P increased to almost 60 % and 30 %, respectively. Under the humid climate of central and western Europe, the % 95P was smaller, but it reached 40 % in western France and the southern UK (Fig. 5c). In the mountainous regions of Norway and in the Alps, i.e., regions with very high mean precipitation and frequent convective rain events, the influence of sporadic ARs was accordingly low. A similar pattern characterized the fractional contribution to the total annual precipitation (% TP, Fig. 5d). Maxima occurred over western France and the western Iberian Peninsula, where ARs accounted for up to 10 % of the total precipitation.

## 3.4 Effect of downscaling

The effect of downscaling was assessed by comparing the 0.75° ERAI reanalysis (Fig. 5, right column) with the 0.22° RCA-ERAI hindcast simulation (Fig. 5, middle column). The lower resolution of ERAI eliminated much of the spatial

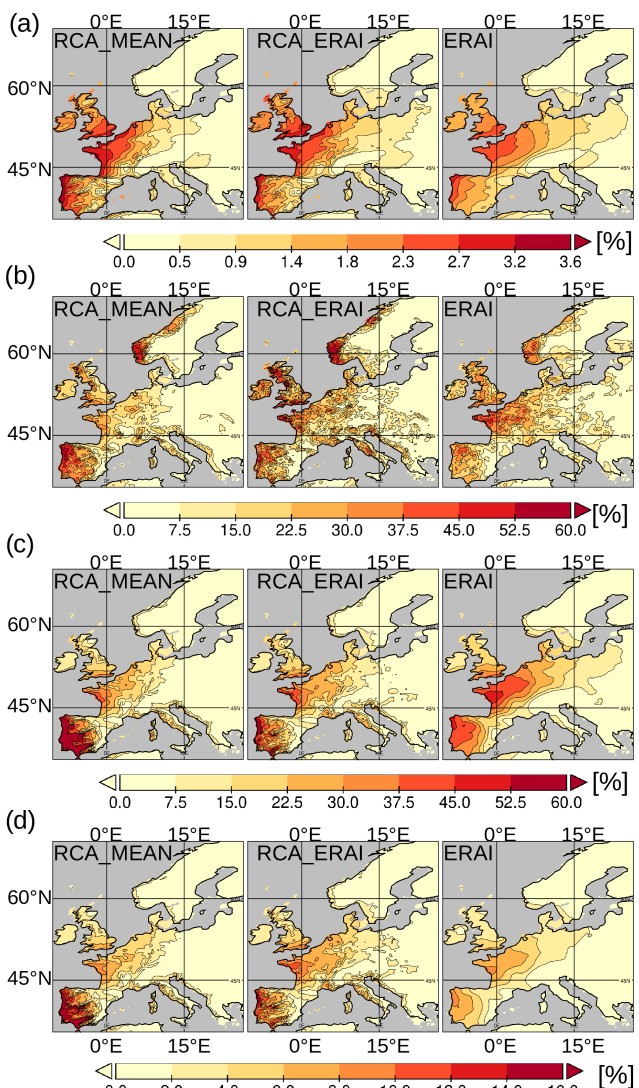

**Figure 5.** ARF expressed as the total number of days a grid cell was covered by an AR during the historical period. **(b)** Percentage of the annual maximum precipitation related to ARs (% AMP). **(c)** Fractional contribution of AR-forced precipitation to the > 95th percentile precipitation (% 95P). **(d)** Same as panel **(c)** but for the total precipitation (% TP).

variability, thus highlighting the effect of the downscaling by RCA. This was most visible in the noisy precipitation-related indices % AMP, % 95P, and % TP (Fig. 5b–d). This was expected because spatial precipitation patterns are modulated by stochastic processes associated with, e.g., small convection cells and further affected by topography. The representation of convection cells and topography has been shown to benefit from a higher resolution (e.g., Feser et al., 2011; Hohenegger et al., 2020; Stevens et al., 2020). The pronounced effect of a higher topographic resolution was seen over the Iberian Peninsula. Here, RCA-ERAI clearly resolved the distinct effect of the prominent west–east topographic features

seen in the fractional contributions to the precipitation budget. Those features appeared as small distinct WSW–ENE bands of alternating high and low $\%95P$ and a $\%TP$ that followed the topographic elements built up by the Sistema Central plateau, the Sierra Morena mountains, and the Penibaetic orogenic system (Fig. 5c and d, and middle columns). By contrast, in ERAI (Fig. 5c and d, right column), there was a simple decline of the $\%95P$, while the $\%TP$ occurred at a longer distance from the coast and did not reflect any topographic imprint. For the Iberian Peninsula, the added value of regional downscaling was previously reported for the simulation of mean precipitation and mean temperature patterns (Gómez-Navarro et al., 2011).

Another noteworthy difference was the considerably larger number of AR-related annual maxima ($\%AMP$, Fig. 5b) over Norway, western Scotland, and Italy in RCA-ERAI than in ERAI. In the latter, the AR influence on $\%95P$ and $\%TP$ was much more extensive in eastern Europe. However, there are no major topographical elevations in eastern Europe, which suggests that resolution is not the only factor accounting for this difference. Rather, differences in the model's physics (e.g., cloud formation) and the surface boundary conditions (e.g., surface temperature, surface roughness parametrization) between RCA-ERAI and ERAI could influence the extent of AR penetration into eastern Europe.

Finally, we note that the comparison between different spatial resolutions might also reflect different noise levels. This noise occurs when isolated grid points located outside the ARs at a given time step exceed the IVT threshold but do not satisfy the geometric and temporal requirements. The different noise levels would then contribute to the total effect of downscaling.

### 3.5 Comparison of the RCA ensemble mean with the ERAI hindcast and ERAI reanalysis

Global climate models can have considerable biases on a regional scale. Consequently, when driven by global climate models at the boundaries, RCA will not perform as well as hindcast models when the driving lateral boundaries are constrained to reanalysis data. Therefore, in the following, before assessing climate change scenarios, we briefly compare the results of the RCA historical ensemble mean (RCA-MEAN, Fig. 5, left column) with the RCA-ERAI hindcast (Fig. 5, middle column) and ERAI reanalysis data (Fig. 5, right column).

For ARF, $\%AMP$, $\%95P$, and $\%TP$, RCA-MEAN reasonably well reproduced the spatial pattern obtained with the RCA-ERAI simulation. The spatial correlation coefficients between RCA-ERAI and RCA-MEAN were 0.98 for ARF (Fig. 5a), 0.86 for $\%AMP$ (Fig. 5b), 0.82 for $\%TP$ (Fig. 5c), and 0.92 for $\%95P$ (Fig. 5d). Similar to RCA-ERAI, the imprint over eastern Europe was distinctly weaker according to RCA-MEAN than according to the ERAI data set.

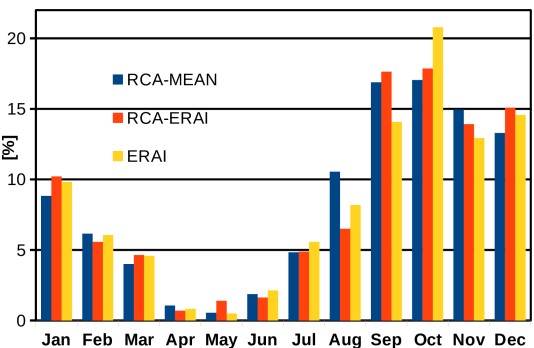

**Figure 6.** Seasonal cycle of detected ARs at $10°W$, expressed as the percent share of the total number of detected ARs. The reference period for the RCA historical ensemble (RCA-MEAN, blue) was 1970–1999, and for the ERAI hindcast (orange) and ERAI (yellow) reanalysis it was 1979–2008.

Pronounced differences between RCA-MEAN and RCA-ERAI also occur over the Iberian Peninsula, where the values of $\%AMP$, $\%95P$, and $\%TP$ are systematically higher in RCA-MEAN. Overall, the best match between RCA-MEAN and RCA-ERAI is for the parameter ARF (Fig. 5a). As ARF is calculated only from the IVT, it is less effected by biases in precipitation. Thus, the main contributor to the biases of RCA-MEAN seen in $\%AMP$, $\%95P$, and $\%TP$ are attributable to the simulated precipitation, not to the IVT. We note that the exact reproduction of precipitation pattern is difficult in coarse-resolution models due to insufficient description in cloud physics and/or the treatment of convective energy (e.g., Prein et al., 2013; Hohenegger et al., 2020).

The detected ARs were also characterized by a strong seasonal cycle that was well reproduced by RCA-MEAN. Figure 6 shows that ARs were most abundant during fall and early winter. The only notable difference was in August, when the relative share was about twice as large in the RCA historical ensemble than in the ERAI hindcast. In addition, RCA-MEAN and RCA-ERAI highly overestimate the number of ARs in September compared to ERAI but underestimate it in October. However, overall, ARs were better represented in the model's climate mode (RCA-MEAN) than in the RCA-ERAI hindcast simulation.

## 4 Future climate change impact on ARs

### 4.1 General response of AR frequency and intensity

The relative change in average moisture transported by ARs for each of the GHG emission scenarios and each of the downscaled global models is summarized in Fig. 7. ARs became consistently more intense; i.e., they had a higher moisture load, in a warmer climate. The average intensity at the end of the century indicated by RCA-MEAN increased by $6\%$ (RCP2.6), $13\%$ (RCP4.5), and $24\%$ (RCP8.5). These

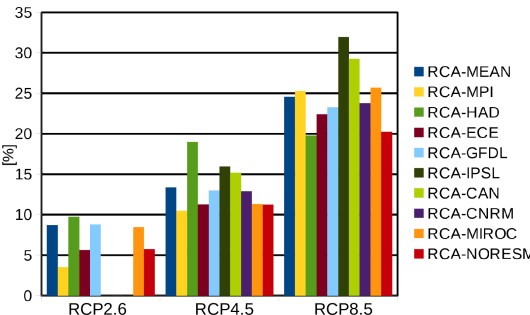

**Figure 7.** Relative change in moisture transport within ARs at the end of the century (2070–2099) compared to the historical period (1970–1999). RCA-MEAN denotes the mean of the individual models. Note that no RCP2.6 realizations are available for RCA-IPSL, RCA-CAN, and RCA-CNRM.

values are generally in line with the corresponding increases in the IVT thresholds (Fig. 2b).

The number of detected ARs also increased (Table 3). For RCA-MEAN, this number increased by 9 %, 19 %, and 24 % in RCP2.6, RCP4.5, and RCP8.5, respectively. These values were roughly proportional to the increase in intensity. However, not only the frequency of ARs but also the spread of the individual realizations at the end of the century increased. The relative change in the ensemble spread (Table 3, second row) increased even more than the ensemble average (RCP2.6 = 17 %, RCP4.5 = 24 %, RCP8.5 = 56 %). This highlighted the large uncertainty with respect to the chosen global model used for downscaling. Advanced approaches for weighted model averaging have been developed to reduce this type of uncertainty and have been tested for ARs occurring over the US (Massoud et al., 2019, 2020; Wootten et al., 2020).

### 4.2 Spatial changes

The change in the spatial patterns is shown in Fig. 8. An overall increase in AR day frequency was determined for scenarios RCP4.5 and RCP8.5 (Fig. 8a). The strongest increase occurred near the Bay of Biscay and adjacent land areas (western France and southern UK). This response over land was more or less consistent across the RCP scenarios but differed in strength, ranging from $\sim +0.2\,\%$ to $> +1.8\,\%$ in RCP4.5 and RCP8.5 (Fig. 8a). This corresponds to a relative increase of 20 %–120 % (RCP4.5) or 40 %–250 % (RCP8.5) over the Iberian Peninsula, the UK, France, Germany, and along the Norwegian coast.

Figure 8b shows the changes in the yearly maximum precipitation attributed to ARs (% AMP). The most robust change was the strong increase over the western central part of Europe, extending from western France along the coast of Belgium, the Netherlands, northern Germany, and Denmark up to the southern coast of Norway. Further sites of stronger AR impact were also visible along the northwestern area of

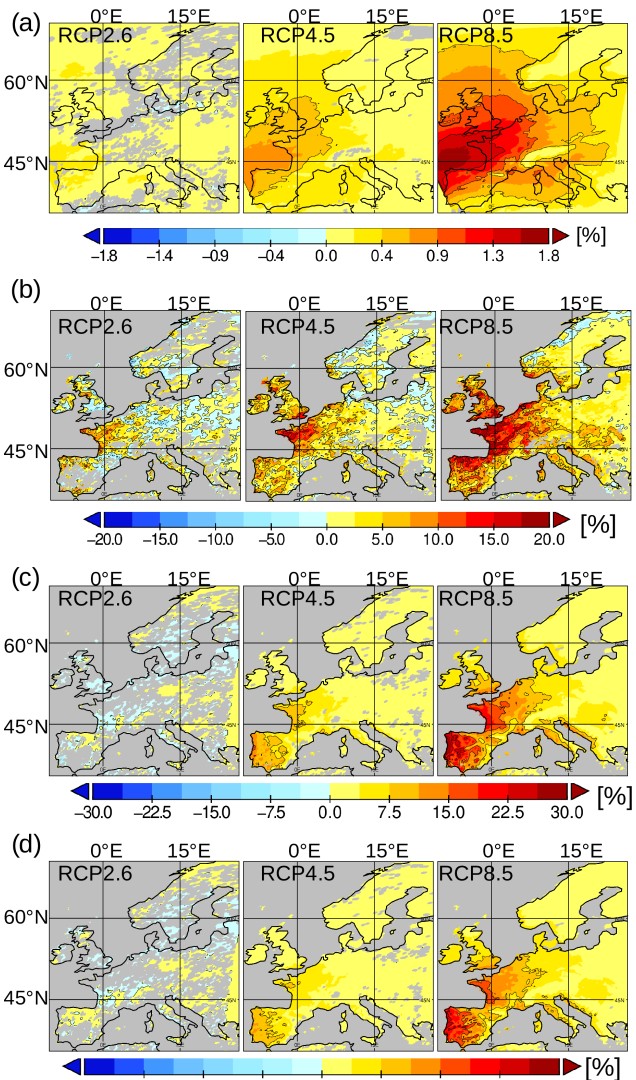

**Figure 8.** Difference between 2070–2099 minus 1970–1999 (i.e., the % values for the future minus the % values for the historical). **(a)** ARF, **(b)** AR-forced yearly maximum precipitation rates (% AMP), and **(c)** change in the AR fractional contribution to the heavy precipitation (% 95P). **(d)** Same as panel **(c)** but for the fraction to the annual total precipitation (% TP). Note that all non-robust changes (at least 66 % of downscaled runs agree on the sign of change) have been masked out. Shown are ensemble mean differences.

the Iberian Peninsula and the southern UK, whereas over southern Scandinavia there were no robust changes. However, in this area, % AMP was already very high during the historical period (Fig. 5b), which limited the potential for further increases. This increase over the UK, France, western Germany, and the Iberian Peninsula was by far the strongest in the unmitigated RCP8.5 scenario. In the moderate scenario (RCP4.5), the changes were less pronounced in eastern central Europe (Germany, Denmark). In the mitigation scenario

(RCP2.6), robust changes were restricted to a small area in NW France (Brittany, Normandy).

The higher AR frequencies and moisture loads also had consequences for local precipitation budgets. The frac-
5 tional contributions of ARs to the heavy-precipitation fraction (%95$P$, Fig. 8c) and to the total annual precipitation (%TP, Fig. 8d) increased nearly everywhere for RCP4.5 and RCP8.5. The most pronounced changes were in regions where the %95$P$ and %TP were already large under histor-
10 ical conditions (Fig. 5c and d). In RCP8.5, the strongest increases primarily occurred in the western Iberian Peninsula and along the French coast (Bay of Biscay), where heavy rain precipitation increased by up to +30% and up to 20%, respectively, compared to the historical period.
The elevated AR fractional contributions %95$P$ and %TP in RCP4.5 and RCP8.5 (Fig. 8c and d) suggested that in these scenarios the increase in the AR-induced precipitation rates was higher than the increase in the average precipitation rates.
The relative changes in mean precipitation and AR-induced precipitation for RCP8.5 are compared in Fig. 9. The mean precipitation change (Fig. 9a) was consistent with the typically dryer conditions over southern Europe and the typically wetter conditions over northern Europe (e.g., Ja-
cob et al., 2014; Kjellström et al., 2018; Teichmann et al., 2018; Gröger et al., 2021a; Christensen et al., 2022 `TS21`). Hence, mean precipitation rates increased only slightly, by at most 12%, over central Europe or even decreased over southern Europe. By contrast, AR-induced precipitation in-
creased ∼ 25%–40% over central Europe (Fig. 9b). Decreasing AR precipitation also occurred in southern Europe but the reductions were weaker than the reductions in the mean rates. The exception was Norway, where AR-induced precipitation decreased while the mean rates increased. The
lower AR precipitation rates well agree with the low response in western Norway of the %AMP, which locally even decreased (Fig. 8b).

## 4.3  Influence of dynamical changes

The dynamic changes were investigated by exploring the
40 route followed by ARs east of the 10° W meridian. In this analysis, AR masks were used (see Fig. 3), and for the model's land grid cells the corresponding latitudinal position where the AR crosses the 10° W meridian was determined. All land grid cells overlain by the respective AR were
45 then flagged with the latitudinal bin at 10° W (for example, 45° N for the bin 45–50° N). The flagged masks were then used to calculate, for every land cell, the percent share for every latitudinal bin (with the sum of all bins at every land point defined as 100%). This was done for the peri-
50 ods 1970–1999 and 2070–2099. Then, the future change (difference between 2070–2099 and 1970–1999) was calculated for RCP8.5. Finally, the latitudinal bins were consolidated

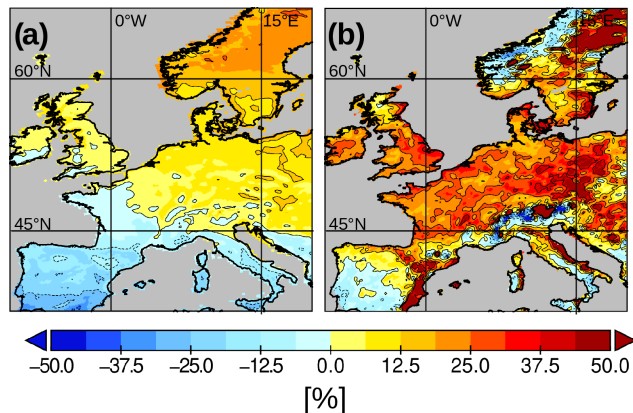

**Figure 9.** Relative change (difference between 2070–2099 and 1970–1999) in **(a)** the average precipitation rates and **(b)** the AR-induced precipitation rates for RCP8.5. Shown are ensemble mean differences.

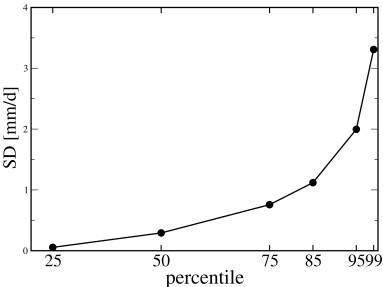

**Figure 10.** Inter-model standard deviation from the nine downscaled models calculated for the 25th, 50th, 75th, 85th, 95th, and 99th percentiles of precipitation. The average over all the land grid cells of the models is shown. The percentiles have been derived from the 6-hourly time series of precipitation.

into three main meridional bands: south of 45 `CE2`, 45–60° N and north of 60° N.

Our analysis showed not only the more frequent occur- 55 rence of ARs in the future at all latitudes but also a change in their composition (with respect to the meridional band where they originate). AR contributions from > 60° N declined everywhere (Fig. 10a), with a ∼ 20% reduction over Norway, but also over many European mountainous regions, such as 60 the Pyrenees, Massif Central, the Alps, and mountains in the Balkans (Fig. 10a). The smaller fraction was paralleled by a relative increase in ARs originating from south of 45° N (Fig. 10c), while the fraction from 45–60° N was more or less unchanged (Fig. 10b). 65

The larger fraction of more southern ARs over Norway (and the smaller fraction from > 60° N) has two implications: (1) the more southern ARs will carry warmer air masses to Scandinavia, such that precipitation will tend to fall more often as rain than as snow. (2) Moisture transport 70 via southern ARs implies that the moisture is routed over longer distances across the central continent before it arrives

Earth Syst. Dynam., 13, 1–20, 2022 https://doi.org/10.5194/esd-13-1-2022

in Norway. ARs traversing the continent can no longer take up significant amounts of moisture but instead lose moisture via precipitation. By contrast, ARs originating from > 60° N and crossing the open North Atlantic can take up moisture until they make landfall at the Norwegian coast. This is in particular the case during the warm season when SSTs are high enough to allow convection or during winter when moisture convergence advects moisture from adjacent areas. These changes likely account for the lower AR precipitation rates determined along the western coast of Norway (Fig. 9b).

## 5 Discussion

### 5.1 Uncertainties with respect to the choice of CMIP5 models

Climate models are designed and validated to simulate the mean state and mean variability of the long-term climate. Hence, the validation of climatic extremes during model development is relatively small. As a result, the models results will differ more for extreme regimes. Figure 11 shows the inter-model standard deviations for the different percentiles of precipitation. The percentiles were calculated for all nine models (Table 2) covering the historical period. For the higher percentiles, the spread over the models clearly increases, such that uncertainties are highest in the extreme precipitation range, i.e., the range where AR precipitation is expected to occur. Therefore, to assess the uncertainties associated with model choice, in the following we consider the spread in the ensemble members with respect to ARF, % AMP, and % 95 P.

Figure 12a TS22 depicts the frequency of AR days. All realizations exhibited a coherent spatial pattern that was similar to the RCA ensemble mean (Fig. 8a), indicating the latter as a representative indicator of the bulk response. The uncertainties in the model spread as indicated by the ensemble's standard deviation (Fig. 12a TS23) were highest in the southern UK, France, Belgium, the Netherlands, Luxembourg, and the western Iberian Peninsula. In conclusion, the uncertainty in the response is highest in those regions where ARs are frequent already during the historical period (Fig. 5a). RCA-HAD, RCA-CAN, RCA-IPSL, and RCA-GFDL showed the strongest response, whereas the response in RCA-MIROC and RCA-ECE was exceptionally weak. The weak climate impact in RCA-MIROC is a direct consequence of low increase in detected ARs (∼ 5 %, Table 3). Only a single model shows a clear signal along the Norwegian coast (RCA-HAD) and whole Scandinavia.

The response of AR-forced annual maximum precipitation events (% AMP) is shown in Fig. 12b TS24. No clear consistent response was detected over western Norway, i.e., the region where in the historical climate the % AMP was highest (Fig. 5b). Some models (RCA-CAN, RCA-MPI, RCA-GFDL, RCA-IPSL) showed distinct locations over Norway

where the impact of ARs was smaller in the future, probably linked to the aforementioned decrease in ARs arising from > 60° N. The most coherent change across the realizations was the relatively strong increase over western France, which in some realizations extended further east. However, also in this region the local ensemble variability was pronounced and in RCA-ECE the % AMP in fact decreased.

Uncertainties with respect to the contribution to heavy precipitation (% 95 P) were highest in France and the Iberian Peninsula (Fig. 12c TS25). High inter-model variation was determined also for the southern tip of the Iberian Peninsula, where the standard deviation was highest. In this region, the contribution of ARs was either reduced, as in RCA-HAD, or increased by as much as 50 %, as in RCA-CNRM. Larger uncertainties were likewise identified for central Europe (France, Belgium, Netherlands, Germany), with the direction of change being consistently positive. In RCA-HAD, RCA-CNRM, RCA-ECE, and RCA-MIROC, the changes over eastern France and Germany were extremely low. A notable decrease, occurring in the UK, was modeled only by RCA-HAD.

In summary, our analysis showed a robust inter-model agreement for the general spatial pattern. However, local uncertainty is high especially in % AMP (i.e., the most severe precipitation events, Fig. 12b), where even the direction of change is not consistent.

### 5.2 Differences to global projections

Our results generally agree with those of the global CMIP5 models (e.g., Lavers et al., 2013; Gao et al., 2016; Ramos et al., 2016), with both indicating more frequent and more intense ARs over Europe. With respect to previous studies, the effect of climate change on AR frequency has been shown to strongly depend on the chosen reference period. Studies that applied the 85th percentile threshold derived from the historical period also to the future period often reported a doubling of the AR frequency (e.g., Lavers et al., 2013; Ramos et al., 2016; Gao et al., 2016). However, as noted above, we calculated separate 85th percentile thresholds for these two periods, and the AR frequency increase was therefore lower (because the IVT thresholds for the future period were higher; Fig. 2) than that in the aforementioned studies but nonetheless 20 %–30 % across the models.

The main differences with respect to global projections occurred over Norway and the Iberian Peninsula, two hotspots of AR impact in Europe. Over the Iberian Peninsula, the distribution of AR-related heavy precipitation was clearly modulated by topographic structures, including the Sistema Central Plateau, the Sierra Morena mountains, and the Penibaetic orogenic system. These valleys and ridges will lead to zonal bands of high and low increases in AR precipitation over the Iberian Peninsula in the future. Over Norway, our regional ensemble did not predict a robust climate change signal for the frequency of ARF, % AMP, % 95 P, or % TP

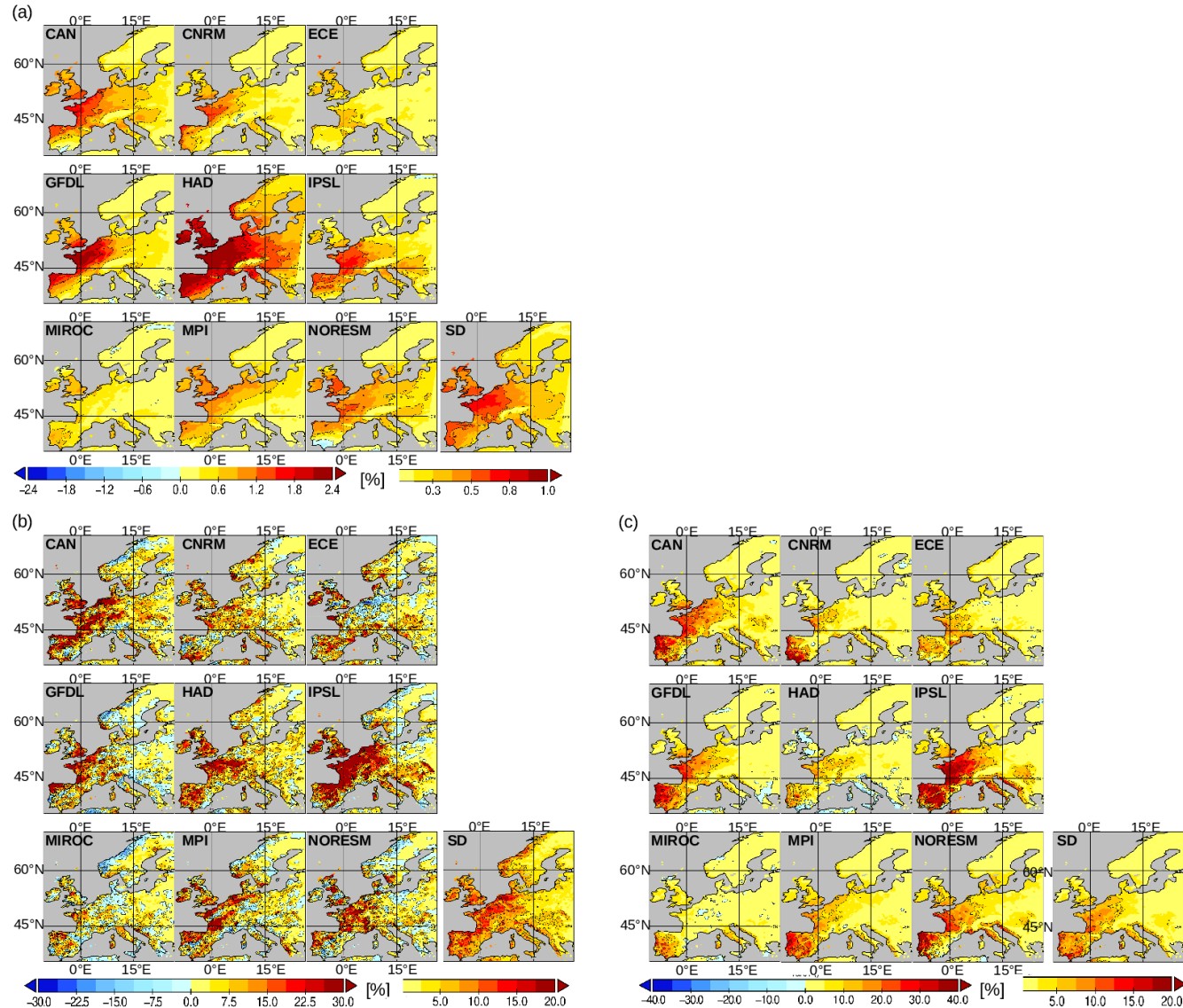

**Figure 11. (a)** Change in AR day frequency (2070–2099 minus 1970–1999). **(b)** Same as panel **(a)** but for percentage in AR-forced yearly maximum precipitation (% AMP); **(c)** same as panel **(a)** but for AR fractional contribution to heavy precipitation (% 95$P$). Note the RCA realization are denoted by their forcing global model simply. SD CE3 denotes the inter-model standard deviation across all nine models (CAN, CNRM, ... NorESM). Long bars indicate the color scale of the ensemble member indices. Short bars are for inter-model standard deviation. All maps reflect changes for RCP8.5.

(Fig. 8). Global CMIP5 models indicated an increase in the regional % TP and % 95$P$ of at least 10 %–20 % according to RCP8.5 (Gao et al., 2016, Fig. 9 therein). In our regional model, % AMP was either negative or positive, depending on the global model, and thus did not yield a clear signal (Fig. 12b TS26 ).

Our finding that in a future high-emission scenario ARs from south of 45° N will be more common over Europe than ARs from > 60° N points to larger-scale atmospheric circulation in the parent global models. While such changes cannot be analyzed in our limited area model, it can be proposed that they are related to changes in the storm track

(Zappa et al., 2013) and/or systematic changes in regional weather systems (Pasquier et al., 2019 TS27 ).

## 6  Summary and conclusions

A high-resolution regional climate model ensemble with a resolution of 0.22° was created to investigate the impact of ARs in Europe. The added value of downscaling was demonstrated by a hindcast that was run to downscale the ERAI reanalysis data set with a resolution of 0.75°. In the central and southern Iberian Peninsula, the contribution of ARs to the regional precipitation budget was shown to be

strongly affected by prominent topographic signatures. This feature was not seen in the ERAI reanalysis data, which instead showed distinct E–W gradients in which precipitation was highest in the west. The AR imprint on the analyzed indices in the ERAI data set was weak over the Iberian Peninsula but stronger in distant parts of eastern Europe compared to the downscaled RCA-ERAI (Fig. 5).

The regional climate model was further used to investigate ARs in present and future climates. Thus, an ensemble of global CMIP5 climate simulations (1.4–3° resolution) was downscaled to reach a 0.22° resolution. In total, 34 simulations were carried out for the GHG emission scenarios RCP2.6, RCP4.5, and RCP8.5.

The historical simulations from the regional climate model ensemble were in good agreement with the global ERAI reanalysis data set and the ERAI simulation hindcast run. In the regional climate ensemble, ARs had the strongest impact in near-coastal regions, explaining up to 60 % of the yearly maximum precipitation rates in regions with orographic uplift (e.g., Norway). Over the Iberian Peninsula and western France, the fractional contribution of ARs to total precipitation (% TP) and heavy precipitation (% 95 $P$) was up to 20 %, and > 40 %, respectively.

Our results showed that in a future warmer climate ARs become more frequent and carry a larger moisture load, consistent with the findings of previous studies (e.g., Lavers et al., 2013; Warner et al., 2015; Gao et al., 2016; Shields and Kiehl, 2016 TS28; Ramos et al., 2016; Shields et al., 2016 TS29, 2019; Massoud et al., 2019; Whan et al., 2020). The potential of ARs to force annual maximum precipitation events is likely to be highest over western France (Brittany) and northernmost Spain, by up to 20 % (RCP8.5), whereas no robust ensemble response was determined over Norway. Our regional high-resolution model thus allows a spatially more accurate calculation of the fractional contributions of ARs to the local water budget than is possible with global Earth system models. It showed that, in the future, the increase in AR-induced precipitation will be larger than the increase in average precipitation such that the fractional contributions of ARs to heavy rain (% 95 $P$) and total rain (% TP) will also be larger, increasing by up to 10 % and 30 %, respectively.

Our study also showed that AR day frequencies over Europe will increase over all latitudes along the 10° W meridian, albeit with a larger increase at southern than at northern latitudes. This leads to a higher fractional contribution of ARs over Europe from more southern latitudes, which in turn affects the route that ARs follow east of 10° W. Since the moisture then travels a longer distance over land, a further uptake of moisture by the AR before it arrives in Norway is prevented. For RCP8.5, this should lead to locally decreased precipitation rates over western Norway. By contrast, during the historical period, a larger share of ARs came from > 60° N, arriving in Norway directly from the North Atlantic.

Elsewhere in Europe, our study clearly demonstrated that, under the higher GHG emission scenarios (RCP4.5 and RCP8.5), a larger imprint of ARs on the regional scale and a larger role of heavy-precipitation forcing events with a potential risk for flooding can be expected. However, under RCP2.6, most of the climate-induced changes are not robust and may thus be responsive to climate mitigation actions.

Our regional assessment of the impact of ARs on Europe must be considered as a first step, since the realized horizontal resolution is still relatively coarse (24 km) and does not explicitly resolve convection. The next generation of regional high-resolution models will improve the resolution to only a few kilometers and allow us to resolve convection (e.g., Giorgi, 2019; Jacob et al., 2020). These advances will allow a more thorough investigation of the processes mediating the response of ARs to climate change and the pathways of ARs across Europe.

**Code and data availability.** The data sets generated during and/or analyzed during the current study are available from the corresponding author upon reasonable request. Numerical model codes are available from the respective literature and corresponding first author. Data to reproduce results presented in this study are available upon reasonable request.

**Author contributions.** . TS30.

**Competing interests.** The contact author has declared that neither they nor their co-authors have any competing interests.

**Disclaimer.** Publisher's note: Copernicus Publications remains neutral with regard to jurisdictional claims in published maps and institutional affiliations.

**Acknowledgements.** The research presented in this study is part of the Baltic Earth program (Earth System Science for the Baltic Sea region; see http://www.baltic.earth, TS31). Regional climate scenario simulations have been conducted on the Linux clusters Krypton, Bi, Triolith, and Tetralith, all operated by the National Supercomputer Centre in Sweden (http://www.nsc.liu.se/, TS32). Resources on Triolith and Tetralith were funded by the Swedish National Infrastructure for Computing (SNIC) (grant nos. SNIC 002/12-25, SNIC 2018/3-280, and SNIC 2019/3-356). The Swedish Civil Contingencies Agency (MSB) and the Swedish Research Council for Sustainable Development (FORMAS) have contributed funding through the HydroHazards project (MSB 2019-0651). Additional funding was obtained by the Swedish Research Council for sustainable development (Formas) through the ClimeMarine project, funded within the framework of the National Research Programme for Climate (grant no. 2017-01949). Finally we thank three anonymous reviewers for thoroughly reading the

manuscript and providing excellent recommendations that helped to improve the article.

**Financial support.** The publication of this article was funded by the Open Access Fund of the Leibniz Association. TS33

**Review statement.** This paper was edited by Gabriele Messori and reviewed by three anonymous referees.

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

## Remarks from the language copy-editor

CE1    Please note the slight edits to the affiliations.
CE2    Please check. What does this 45 refer to?
CE3    Please note that "SD" is our house standard. Should this also be updated in Table 3?

## Remarks from the typesetter

TS1    If possible, please give deceased date.
TS2    If possible, please give deceased date.
TS3    The composition of Figs. 2–5 and 8–11 has been adjusted to our standards.
TS4    Zhu et al. (1998) changed to Zhu and Newell (1998). Please confirm.
TS5    Gimeno et al. (2018) is missing in the reference list. Please check.
TS6    Nayak et al. (2017) changed to Nayak and Villarini (2017). Please confirm.
TS7    Alfieri et al. (2016) changed to Alfieri et al. (2017). Please confirm.
TS8    Is this Shields and Kiehl (2016a), Shields and Kiehl (2016b) or both? Please check.
TS9    Is this Dietrich et al. (2019a), Dietrich et al. (2019b) or both? Please check.
TS10    Is this Dietrich et al. (2019a), Dietrich et al. (2019b) or both? Please check.
TS11    Is this Dietrich et al. (2019a), Dietrich et al. (2019b) or both? Please check.
TS12    Thomson et al. (2011) is missing in the reference list. Please check.
TS13    Nayak and Villarini (2016) changed to Nayak and Villarini (2017). Please confirm.
TS14    Lavers and Villarini (2012) changed to Lavers and Villarini (2013). Please confirm.
TS15    Lavers and Villarini (2012) changed to Lavers and Villarini (2013). Please confirm.
TS16    "UTC" inserted. Please confirm.
TS17    Is the end of the enumeration here? Please check.
TS18    "UTC" inserted. Please confirm.
TS19    Davies (2013) changed to Davies (2014). Please confirm.
TS20    Thandlam et al. (2021) changed to Thandlam et al. (2022). Please confirm.
TS21    Christensen et al. (2021) changed to Christensen et al. (2022). Please confirm.
TS22    There is no Fig. 12 submitted. Please check numbering.
TS23    There is no Fig. 12 submitted. Please check numbering.
TS24    There is no Fig. 12 submitted. Please check numbering.
TS25    There is no Fig. 12 submitted. Please check numbering.
TS26    There is no Fig. 12 submitted. Please check numbering.
TS27    Pasquier et al. (2018) changed to Pasquier et al. (2019). Please confirm.
TS28    Is this Shields and Kiehl (2016a), Shields and Kiehl (2016b) or both? Please check.
TS29    Shields et al. (2016) is missing in the reference list. Please check.
TS30    Please note that the section "Author contributions" is mandatory. Please provide the text for this section in complete sentences. Please see https://publications.copernicus.org/for_authors/obligations_for_authors.html for more information
TS31    Please provide last access date.
TS32    Pleas provide last access date.
TS33    Please note that there is a discrepancy between funding information provided by you in the acknowledgements and the funding information you indicated during manuscript registration, which we used to create this section. Please double-check your acknowledgements to see whether repeated information can be removed from the acknowledgements or changed accordingly. If further funders should be added to this section, please provide the funder names and the grant numbers. Thanks.
TS34    Please ensure that any data sets and software codes used in this work are properly cited in the text and included in this reference list. Thereby, please keep our reference style in mind, including creators, titles, publisher/repository, persistent identifier, and publication year. Regarding the publisher/repository, please add "[data set]" or "[code]" to the entry (e.g. Zenodo [code]).
TS35    Update inserted. Please confirm.
TS36    Cabos et al. (2020) is not mentioned in this paper. Please check.
TS37    Please give more information like DOI number or URL and last access date.
TS38    Dettinger (2011) is not mentioned in this paper. Please check.
TS39    Please check page range.
TS40    Di Luca et al. (2012) is not mentioned in this paper. Please check.

TS41 Emori and Brwon (2005) is not mentioned in this paper. Please check.

TS42 Please provide page range or article number.

TS43 Please provide page range or article number.

TS44 If possible, please provide an update.

TS45 Hagemann and Dümenil (1998) is not mentioned in this paper. Please check.

TS46 Hagemann et al. (2009) is not mentioned in this paper. Please check.

TS47 Harvey et al. (2020) is not mentioned in this paper. Please check.

TS48 Please provide volume and check page range.

TS49 Ho-Hagemann et al. (2017) is not mentioned in this paper. Please check.

TS50 Please provide page range or article number.

TS51 Huang et al. (2020) is not mentioned in this paper. Please check.

TS52 Jacob (2001) is not mentioned in this paper. Please check.

TS53 Pleas provide all author names.

TS54 Please provide all author names.

TS55 Please provide volume and page range or article number.

TS56 Jeworrek et al. (2017) is not mentioned in this paper. Please check.

TS57 Jungclaus et al. (2013) is not mentioned in this paper. Please check.

TS58 Please provide volume and page range or article number.

TS59 Kaiser-Weiss et al. (2019) is not mentioned in this paper. Please check.

TS60 Kelemen et al. (2019) is not mentioned in this paper. Please check.

TS61 Please provide all author names.

TS62 Please provide last access date.

TS63 Lavers and Villarini (2015) is not mentioned in this paper. Please check.

TS64 Marsland et al. (2002) is not mentioned in this paper. Please check.

TS65 Please provide all author names.

TS66 Please check page range.

TS67 Please provide intitial(s) of last author.

TS68 Neiman et al. (2008) is not mentioned in this paper. Please check.

TS69 Payne et al. (2020) is not mentioned in this paper. Please check.

TS70 Update inserted. Please confirm.

TS71 Primo et al. (2019) is not mentioned in this paper. Please check.

TS72 Please confirm change.

TS73 Ralph and Dettinger (2011) is not mentioned in this paper. Please check.

TS74 Please provide page range or article number.

TS75 Please give more information like DOI number or URL and last access date.

TS76 Schiemann et al. (2018) is not mentioned in this paper. Please check.

TS77 Sein et al. (2015) is not mentioned in this paper. Please check.

TS78 Please provide volume and page range or article number.

TS79 Sein et al. (2020) is not mentioned in this paper. Please check.

TS80 Please provide all author names.

TS81 Sodemann and Stohl (2013) is not mentioned in this paper. Please check.

TS82 Soto-Navarro et al. (2020) is not mentioned in this paper. Please check.

TS83 Please provide all authro names.

TS84 Please give more information like DOI number or URL and last access date.

TS85 Vancoppenolle et al. (2008) is not mentioned in this paper. Please check.

TS86 van Haren et al. (2015) is not mentioned in this paper. Please check.

TS87 Please provide volume and page range or article number.

TS88 Please check DOI number.

TS89 Please provide all author names.

TS90 Please provide page range or article number.

TS91 Please provide page range or article number.

TS92 Wang et al. (2017) is not mentioned in this paper. Please check.

TS93 Please provide volume and page range or article number.

TS94 Zhu et al. (2020) is not mentioned in this paper. Please check.

TS95 Please check DOI number.

Earth Syst. Dynam., 13, 1–20, 2022 https://doi.org/10.5194/esd-13-1-2022