# Peer review of "Atmospheric rivers in CMIP5 climate ensembles downscaled with a high resolution regional climate model"

_Earth System Dynamics, 2021_

## Author Comment (AC2)

**This paper investigates the behavior of North Atlantic ARs in reanalysis and ensembles of global simulations in historical and future simulations. The manuscript is in good shape. It is well-written and has a clear structure. I don't have major concerns for the paper, but some minor suggestions and comments for discussion:**

We thank the reviewer for a thorough review of our manuscript and his/her valuable suggestions that really help to improve the manuscript.

**L193: Guan and Waliser (2015) does not provide a thorough review of AR detection methods, instead, GW15 explained their AR detection algorithm in detail. Since GW15's algorithm is not adopted in this paper, it is better to cite a different paper as the "overview". For example:**

**Shields et al. (2018): Atmospheric River Tracking Method Intercomparison Project (ARTMIP): project goals and experimental design, Geosci. Model Dev., 11, 2455-2474, https://doi.org/10.5194/gmd-11-2455-2018, 2018.**

Thank you for this suggestions. We will cite Shields et al., 2018 and remove Guan and Waliser (2015).

**Figure 4: I wonder how the duration distribution would change if the duration threshold to tuned to 6 hours or 12 hours. How will the duration (including those excluded short-lived events (<18hrs)) change in the future climate?**

We have have recalculated this exemplary for the MPI-ESM realization. The main effect is, as expected, an increase of about 55% in the total of detected ARs. The left hand figure below compares the distributions calculated with a threshold set to 18 hours (orange) and 12 hours (blue). Generally, both distributions are similar and indicate most ARs in the short term fraction (with >50% ARs 30 hours or shorter). The right hand figure displays the relative change for the RCP8.5 scenario [%, i.e. future minus historical period *100 / historical period)]. In both distributions the tendency towards extra long (48 hours or higher) ARs can be seen. However, these extended ARs remain typically rare (<10%) in both historical and future climate.

[Figure]

**Figure 5a: instead of showing the actual number of AR days, it might be helpful to show the percentage of AR days to the number of days of the period.**

We agree. We have adopted figure 5a accordingly. We have also changed Fig. 8 in the same way which displays the change in AR days in a future warmer climate. (See the new figure 5 in our reply to RC3)

**Table 3 and L367: Is it possible that the dynamical field is more active in the future simulation as well?**

**Table 3 shows that the number of AR increased in the future runs: it makes sense if the historical AR threshold is applied in the future run – however, here the future ARs are detected with IVT thresholds calculated from future simulation, so the higher moisture load is reflected in the future IVT threshold to**

**some extent. Therefore, I am curious if the change in dynamical filed also contribute to the increase in future AR number.**

Yes, it is likely that both dynamical as well the thermodynamical changes contribute to the increase in AR number. Hence, it could be that more ARs are initiated in the source region (outside our model domain) or that the pathway from the source area to Europe changed. This we can not analyze from our results but this question was addressed in global CMIP5 models (which are more suited for this question) . Lavers et al., 2013 argue for mostly thermodynamical changes (moisture increase) for the future increase in ARs. Gao et al., (2016), likewise found thermodynamics playing a dominant role but pointed out potential dynamical changes in the seasonal distribution of ARs.

**L450: I am not sure if I fully understand the methodology. Are the origins of ARs being categorized by the southernmost latitude? If so, isn't it more related to the curvature of AR's shape than the actual origin?**

No it's not the southernmost. We will reformulate this description. First, for every AR that is detected over land we check the time and the latitude when the AR was registered for the first time at 10°W. Hence if we have an AR entering a land grid cell we "look" in our "archived" when this AR it was registered for the first time. For the post-statistical analysis we then summarize by regions south of 45° degree between 45-60° and >60° to obtain the results.

**L563: please rewrite, grammar problem.**

"The regional climate model is further used to investigate the dynamics of ARs in present and future climate. For this in an ensemble of in total 34 simulations was carried to downscale global climate model scenarios from the CMIP5 suite with a coarse resolution between 1.4 – 3°. The models were used to downscale three greenhouse gas scenarios (RCP2.6, RCP4.5, and RCP8.5). "

Done. The paragraph reads now:

The regional climate model is further used to investigate ARs in the present and future climate. For this an ensemble of global CMIP5 climate simulations (1.4° - 3° resolution) was downscaled to reach 0.22° resolution. In total 34 simulations were carried out with the regional model following greenhouse gas scenarios RCP2.6, RCP4.5, and RCP8.5. For this in an ensemble of in total 34 simulations was carried to downscale global climate model scenarios from the CMIP5 suite with a coarse resolution between 1.4 – 3°. The models were used to downscale three greenhouse gas scenarios (RCP2.6, RCP4.5, and RCP8.5).

---

## Author Comment (AC3)

**The authors make use of downscaled versions of global climate simulations from the CMIP5 ensemble to study the atmospheric rivers reaching Europe in the present and future climates. They find that ARs will become more frequent and stronger in the future, especially in the RCP8.5 scenario. The results also show that the orientation of ARs will change with for example more ARs coming from the south reaching Scandinavia (Norway).**

**The study is interesting but I got confused with some of the fields presented in the figures, which quality is not great. Some information about the methods and the fields displayed in the figures is missing, the methodology and results are not well discussed and compared to previous studies, and there are many technical mistakes (typos, English, missing words). Therefore, I think the manuscript needs major revision in order to be in a publishable state.**

We thank the reviewer for a thorough review of our manuscript and his/her valuable suggestions that really help to improve the manuscript. We regret that we were too short in the description of the methods and other parts of the text. We will be more comprehensive in a revised version and give lacking details.

**Major comments:**

**About the methods:**

**- The temporal scale of the RCA model outputs is not clear. From line 209, it seems to be 6-hourly but it would be great to also mention it in section 2.1 or in section 2.2. In the same line, is the extreme precipitation determined from 6-hourly or daily data?**

RCA outputs the specific humidity at 6-hourly output intervals. The precipitation fields represent  accumulated values over the 6-hourly output period. We will note the temporal scale of the used fields more prominently in an own paragraph section 2.1.

**- The authors use an AR minimum length threshold of 1500 km. This is quite short compared to previous studies and I guess the reason for this low value is the limited extension of the domain to the west. Can the authors state the reason for such threshold and maybe discuss it in light of previous studies?**

In this study we followed the previous approach from Lavers and Villarini (2013) who used a minimum length of 1500 km to detect ARs in an ERA-I reanalysis product. However, you're completely right that due to our limited domain the algorithm does not detect

1. ARs that do not reach Europe but remain outside over the Atlantic Ocean. Thus, our study can not be compared 100% to global CMIP studies on ARs that take into account also those ARs.

2. Over the western Iberian Peninsula, which is located relatively close to the western model boundary, some AR could be missed or detected with delay (as it may take longer to reach the 1500 km criterion). Over the UK and Norway this does not play a significant role as these countries lie far away from the models lateral boundary.

Thank you for that comment. We will include a short paragraph that makes this clear in a revised manuscript.

**- Can the authors confirm that only one AR is detected at every timestep? What happens when an AR covers two latitudinal bins and/or exceeds the IVT 85th percentile in two adjacent 5º bins?**

Yes, we can clearly confirm this, as it is checked by our algorithm. ARs are detected separately for each of the 5° latitudinal bins. At this stage it is indeed possible that one AR time step is recorded twice. Therefore, after detection, the whole record is checked for double AR time steps (year,  month, day, and hours information is saved). All double time steps are removed. We will make this clear in a revised version.

**- Do the authors define a mask of every AR to link them to precipitation?**

Exactly, for every single AR a mask array is created which contains likewise information about the exact date and time. All this in 6-hourly resolution.

**- Lines 223-224: I think the authors should emphasize the fact that the AR detection threshold is different between the present and future periods in contrast to previous studies and what the advantage of this method is. As for now, it is written later in the manuscript (section 5.2) but it should appear upfront. I believe that this choice limits the influence of the larger moisture content in the atmosphere in the future climate on the results. Is that true? Can this aspect be discussed if relevant?**

We also think this fact should be emphasized more prominently already in the methods section.

As stated in Lavers et al., (2012; 2013) the $85^{th}$ percentile of all 12:00 (noon) time steps represents approximately the median value of moisture content within real observed ARs in todays climate (1998-2005). Thus, our approach conserves the relationship between the median moisture content of ARs with the $85^{th}$ percentile of all noon time steps. The other way to do would be to apply the historical threshold also to the future atmosphere which contains much more moisture also in the background field. This emphasizes more the thermodynamical aspect but does not ensure full compliance with the algorithm developed for present day AR characteristics.

We will add a new paragraph that discusses this.

**- I believe the ERA-Interim reanalysis is not really described. For example, the spatial and temporal resolution used in the study is missing.**

Yes, the information on temporal resolution is missing. Spatial resolution (0.75°) is somewhat hidden (section 3.2.3). We will provide this information more prominently and comprehensive in the methods section.

**- In the caption of Fig. 2 and line 329, it is written that ARs are tracked. However, the AR tracking is not explained. I suspect the tracking is used to check the AR persistence and involves ARs masks. Therefore, please add this information in section 2.3.**

Yes, the word tracking is misleading here (as it suggests a kind of Lagrangian/Eulerian approach) and we will replace it by "detection". We did not really track ARs but, as you said, checked persistence of ARs.

**About Section 3.2.3:**

**Lines 351-354: Can the authors explain a bit more how a negative temperature bias in the regional climate model is linked to a "too high moisture load" and to the higher number of ARs in September in the hindcast compared to ERA-Interim? I would expect a higher temperature to be linked to more moisture. Moreover, wouldn't it be more useful to look at the precipitation in the hindcast, in ERA-Interim, and in the E-OBS dataset, instead of looking at the temperature? Could it also be useful to assess the difference in the specific humidity between the hindcast and ERA-Interim?**

Sorry that was not well explained. We consider here the different mean climates outside the model domain (global GCM) and inside RCA which have different thermodynamic equilibrium states. Our line of argumentation was: if RCA has a cool bias (compared to ERAI outside) then we assume that the moisture content of inflowing warmer air masses is higher than the moisture content in air masses within RCA. Then, an increased inflow of the warmer and more humid air with the beginning of fall from outside RCA would probably result in an increase in incidents where the moisture content exceeds the $85^{th}$ threshold that is based on RCA thermodynamics. However, this is a bit speculative without further analysis which goes beyond the present paper. Therefore, we will remove this paragraph as it is not important for the main conclusions of the paper.

We think indeed it's a good idea to show the comparison in specific humidity between ERAI and our ERAI hindcast. Unfortunately the EOBS data set does not provide the this variable, so we have to exclude EOBS in this comparison.

**Lines 354-356: This sentence must be rewritten. It is not clear at all.**

We agree and will remove this section as it is too speculative (see previous comment above).

**What would be the impact of using the E-OBS precipitation instead of ERA-Interim's in the results displayed in the right column of Fig. 5b-d?**

We recalculated figures 5b-d using the AR masks from ERA-I combined with precipitation from the E-OBS dataset. We note, that by doing so, we violate the IVT equation shown in line 200 which means there is no physical consistency between the atmospheric moisture content and precipitation. However, the result is shown in the right-most column. For the percentage of yearly maximum precipitation related to ARs (see below figure 5b) the spatial pattern is more or less the same. Differences occur over Norway and the UK where ERAI_eobs_rain shows a weaker signal compared to ERAI.

With respect to the contribution of AR forced precipitation to the total >95th percentile precipitation (figure 5c) and the AR forced precipitation to the total amount of precipitation (figure 5d) we note that the pattern is the same as for ERAI (third column) but the amount of rain related to ARs is overall higher compared to ERAI.

[Figure]

*Altered Fig. 5 from the submitted version. AR frequency expressed as % of AR days during the historical period. b) Percentage of annual maximum precipitation related to ARs .c) contribution of AR forced precipitation to the >95th percentile precipitation fraction. d) same as c) but for the total precipitation.(Note that in a) the unit has been changed from total number AR days within 30 years to %AR days of total days as recommended by RC2).*

**About section 5.2:**

**Gao et al. (2016) showed that ARs became more frequent north because of the poleward shift of the eddy-driven jet. That is not what the authors seem to obtain by distinguishing the "origin" of the ARs. I believe it would be more interesting and useful to look at the eddy-driven jet response in all scenarios to explain the responses in the number of days with ARs and AR-forced precipitation?**

That's true. Gao et al. found in CMIP5 models a peak in AR frequency between 45-55°N. Indeed, our Fig 8a indicates clearly that this peak is farther to the south compared to Gao et al., 2016). This suggests that the regional model may systematically steer ARs on a more southward path towards Europe compared to the global models used in Gao. We think this is an interesting result and will discuss this in a revised version.

Though it would be interesting to look on the eddy driven jet this would require further intense analysis. Also we think our regional model is not an ideal tool for this analysis as it excludes wide areas of the North Atlantic.

**Remove sentences "At least… (Ma et al. 2020). The authors…likely reasons." on lines 546-548 because it is not relevant for the present discussion.**

We agree and will remove the sentences.

**About the figures' content:**

**- Figure 2: is the 85th percentile determined using all points within the 5º bins and all time steps? In any case, please state somewhere how the values shown in this figure were calculated.**

Yes, it was calculated from all the grid cells in the range of the respective 5° bin and all time steps. We will make this more clear in a revised version.

**- Figure 4, 7: how is the "average moisture transport over land" calculated? Are the authors using a mask for the ARs and averaging the IVT within the mask? Please explain.**

Yes this is exactly the way it was done. We will include this explanation in a revised version.

**- Figure 5: Row a): the authors describe this row as AR frequency in the caption but, as written in the text, it is only a number of days. One would have to divide by the total number of days in the period to get a frequency. Moreover, since the model output is 6-hourly, I do not understand how the authors convert it to a number of days. What if an AR covers two days during its lifetime? Is it counted twice? Also, if the AR lasts a minimum of three time steps (because of the 18h minimum duration) in the same day, it is counted only once, correct?**

Sorry, we were sloppy with describing the method. For the analysis we classified a certain day as AR day if within the 24 hours at least one AR incident was recognized. That means even if only one of the four 6 hour time steps during the day was impacted by an AR, this day is counted as AR day. We will make this clear in a revised version. Consequently, an 18 hour-lasting AR that covers two subsequent days is counted as 2 days. We will also calculate the frequency as suggested (here in % AR days in the period as suggested by reviewer RC2).

**Rows b), c), and d): I find not clear what is shown in those panels. It is worth mentioning in the text how those "indices" are calculated and keep the same names throughout the manuscript. Can the authors explain why ARs contribute to the yearly maximum (row b) over southwestern Norway or northern UK but barely contribute to the extreme precipitation (row c)?**

We regret the confusion that our description of the indices caused. We will exactly explain how the indices are calculated in a revised version (see below).

**Row b): I understand this figure as the percentage of years (among the 30 years of the period) for which the maximum precipitation occurs when there is an AR. Is that correct? In any case, the text should be clarified. The same comment applies to the rows c) and d).**

Exactly, if in 15 years out of the 30 year period the annual maximum precipitation can be related to an AR we get a value of 50%.

In figure c) we

1. sum up the accumulated precipitation volume that occurs in all rain events that exceed the 95th percentile rain rate.

2. sum up the accumulated precipitation volume that occurs in all rain events that exceed the 95th percentile rain rate and can be related to ARs

3. we calculated how big is the fraction (%) of the sum in step 2 in the sum calculated in step 1.

In figure d)

we do same. But instead the >95 th percentile we consider the all rain events (not only the >95th percentile events)

This was done in the same way as done in GAO et al., (2016). See figures 8 & 9 therein. Gao et al. denoted this fractional "fractional contribution of ARs to total precipitation or TO >95th percentile precipitation". We will do the same naming in a revised version.

**Row c): how do the authors relate the low values for the Norwegian coast to the study of Benedict et al. (2019) who found that 85% of the extreme precipitation events are related to ARs?**

**Benedict, I., K. Ødemark, T. Nipen, and R. Moore (2019): Large-Scale Flow Patterns Associated with Extreme Precipitation and Atmospheric Rivers over Norway. Mon. Wea. Rev., 147, 1415-1428. https://doi.org/10.1175/MWR-D-18-0362.1**

If we understand Benedict et al., (2019) right, then they calculated the percentage of the number (N) of >99th percentile rain events. This is not what we did. We followed Gao et al. 2016 and calculated the accumulated volume of precipitation (not the number of events; see above). In the below figure we applied the approach of Benedict et al. (2016) to our RCA_ERAI run. As can be seen for southwestern Norway up to 80 % of >99 percentile events are related to ARs which is comparable with Benedict et al., (2019).

[Figure]

*% of number of AR related >99th percentile rain events to total number of >99th percentile rain events (as done in Benedict et al., 2019).*

**- Figure 8: is this figure showing a simple difference between the future and the historical experiments or does it show a relative change?**

Figure 8a (AR days) shows a simple difference; not the relative change. Likewise, for figures 8b, 8c, 8d) and 8e) simple differences were shown (i.e. the % values for the future minus the % values for the historical). We will make this clear in revised version.

**Caption: I would not call what is displayed "climatological indices". Please remove. Panel b) does not show "precipitation rates" if it is similar to Fig. 5. Please use the same wording for Fig. 8 as for Fig. 5.**

We agree. We will fully harmonize figure 5 and 8 in a potential revised version

**"Note all non-robust" -> Note that all non-robust**

We will change this in a potential new version

**What is the difference between panels c and d? One of the two is not shown in Fig. 5.**

Figure 8c is showing the change in % of number of events in the 95[th] percentile (similar to Benedict et al., 2016 but for the 95[th] percentile instead of the >99[th] percentile fraction). Figure 8d is the % of accumulated rainfall associated with AR events in the >95[th] percentile fraction, hence it is the future change to what is shown in 5 c

**Why is the unit in panel c is % if the what is shown is "the number of of events" as written on lines 400-401?**

It is the difference change in number of events. Hence, % number of AR events in >95[th] percentile fraction in future climate MINUS % number of AR events in >95[th] percentile in historical climate. However, we will remove figure 8c as it does not bring up added information to what is shown in 8d. Then, Figure 8 is will be completely consistent with Figure 5.

**- Figure 10: Are the authors sure that panel a) is for ARs originating north of 60ºN and panel b) for ARs originating south of 45ºN? It does not seem in agreement with the text. For example, the sentence "the RCA ensemble clearly shows a relative increase of those ARs originating south of 45ºN (Fig. 10a)." However, Fig. 10a only shows negative values and the caption says that panel a) is for ARs originating north of 60ºN. Please make sure that the caption and text (from line 458 to 475) correspond to the figure. Can the category 45-60ºN be displayed as well? In the caption, what does "relative contribution" mean? Does it show the relative difference between the future and historical experiments or is it a simple difference?**

This is a mistake. It should be

"the RCA ensemble clearly shows a relative increase of those ARs originating south of 45ºN **(Fig. 10b)**." (Figure 10b shows increase of originating ARs south of 45°.)

Below we replotted Figure 10 with the fraction 45-60 °N included. The changes are everywhere below 3 %.

[Figure]

*Bin-analysis of AR detection along 10°W. a) Fraction of AR occurrences caused by ARs that were detected south of 45°N at 10°W. b) same as a) but for the fraction 45-60°N. C) same as a but for ARs originating from south of 45°N. Shown is the*

*change for RCP8.5 (2070 - 2099 minus 1970-1999). This figure reads like: In southern Norway the AR fraction from south of 45°N (and at 10°W) reduced by up to 20% in the future climate compared to the historical climate.*

In general the figure reflects the fact, that the relative increase in registered ARs along 10°W increases stronger in the south than in the North (see also Figure 8a).

**- Figure 11: Why does the color bar for the standard deviation panels exhibit negative values? A standard deviation is positive. Is the STD panel a difference between the future and historical standard deviation or the standard deviation of the responses displayed?**

The standard deviation denotes the inter-model spread  of the responses. The negative values arise from an automated scaling error in our plotting package. We will remove the  negative value from the color bar.

**About the figures' quality:**

**I think the quality of the figures should be improved.**

**Figures 2, 3, 4, 8, 9, 10, and 11 exhibit gray frames around the panels and around the color bars. Could they be removed?**

**Figure 9 exhibits a colored line in between the two panels as if another figure was below.**

**Figures 1, 5, 9, and 10 exhibit weird coastlines over Greece. Moreover, Crete and the Balearic Islands are missing.**

**Figure 11 has too small labels for the latitudes (longitudes missing) and for the color bars. Moreover, the coastlines are discontinued at 20ºE.**

**Figures 4, 6, and 7: the frames, tick marks, and background grids are almost invisible. Please make them darker or black.**

**Figures 7, and 11, and Tables 2 and 3: Please arrange the GCMs in alphabetical order as the authors did for Fig. 2.**

**It would be great if all figures showed the same domain.**

**Figure 8: can the columns be rearranged such that RCP2.6 is on the left and RCP8.5 on the right? I find it more intuitive and it would be consistent with Figs. 2 and  7 and Tables 2 and 3.**

**Figure 2: the gray and yellow lines are barely visible.**

**Figure 5: In the left column, panels b), c), and d) should have RCA_MEAN as title instead of RCA_ENSM. Better remove the titles of the three bottom rows as it would make the panels bigger, improving their readability.**

In case a resubmitting is encouraged, we will replot all the figures taking into account the above recommendations together with those from the other reviewers.

**Minor comments:**

**Lines 25-28: it seems from this sentence that Norway is in Central Europe. Please rewrite this sentence, maybe splitting it in two.**

We agree.

**Sometimes, the authors write ERA-I and sometimes ERAI. Please be consistent in the text and  captions and make sure the same acronym is used everywhere (I suggest ERAI).**

We agree.

**Line 138: why is the coupling between the RCA model and NEMO only over the North and Baltic seas? What about the Mediterranean and the Norwegian seas?**

The regional setup of NEMO was originally developed for the North Sea and Baltic Sea which need a very high grid resolution (~3.7 km on average, and 56 vertical levels). Hence NEMO needs very high computational resources. Therefore the other sea had to be excluded.

**Line 151: "in a huge ensemble": to which ensemble do the authors refer to? Does it have a name? This ensemble seems similar to the one used in the present study so why not using it?**

Yes it's the same ensemble. Will will make this clear.

**Caption of Figure 1: Please rewrite it. This figure mainly shows the topography of the domain in brown and the bathymetry in blue-green colors.**

We will change the caption appropriately.

**The authors very often use very "greenhouse gas scenarios". I suggest to use instead "greenhouse gas emission scenarios" or probably better "GHG emission scenarios".**

We agree.

**Lines 202-203: please add the units of g, the wind, and dp.**

Will be included.

**Line 203: "In the two hydrostatic models…": I assume that the authors here refer to the regional climate model RCA and to the IFS model. Why does the reader need this information?**

Yes, the information is obvious from the equation, so we can remove the sentence.

**Caption of Fig.2 : "IVT thresholds" -> 85th percentile of IVT**

will be changed.

**"…for the ensembles' historical…" -> … for all models and the historical…**

will be changed.

**Line 278 (page 10): "heavy precipitation": please write that this is defined using the 95th percentile in order to relate it to what is shown in Fig. 5c.**

Will be done.

**Lines 279-280: What are "the mean climatic conditions in Europe"? Moreover, it rains a lot in Southwestern Norway and that is not reflected in Fig. 5a.**

What we basically want to express is: in humid regions with plenty of rain (both in the total and >95$^{th}$ percentile fraction) ARs can not contribute to the total rain as they do in dryer regions (like Iberia). We will either rephrase this or remove the part of the sentence.

**Line 296: What is meant with "weather regimes"? Moreover, what is the point of lines 295-298?**

With weather regimes we mean frequently re-occuring  synoptical weather patterns such as investigated in Pasquier et al., 2020). We will include a reference to the study of Pasquier at this place.

We wanted to make clear that individual ARs in historical simulations of global climate models can not be directly compared to ARs in hindcast simulations. May be not all readers are aware of this.

**Line 321: Do you mean effect of the downscaling? With "regionalization", it sounds to me like the authors split the domain into different regions.**

We used the two term synonymous. We will replace "regionalization" by "downscaling" in a next version

**Line 324: Can the authors explain the "factor of 10"?**

The factor refers to the grid cell size in RCA (550-600 km$^2$) and the ERAI reanlysis (~6000 km$^2$)

**Line 378: "AR" -> ARs**

will be changed.

**The reference Massoud et al. (2020) does not seem appropriate here as the paper is not about the US but about the Middle East. Maybe the authors meant the following reference:**

**Massoud, E. C., H., Lee, P. B., Gibson, P., Loikith, and D. E., Waliser (2020): Bayesian Model Averaging of Climate Model Projections Constrained by Precipitation Observations over the Contiguous United States, Journal of Hydrometeorology, 21, 2401-2418. https://doi.org/10.1175/JHM-D-19-0258.1**

We will check and change accordingly. Thank you for the hint.

**Lines 418-419: what is meant with "contribution anomalies"? Could it be replaced by responses?**

Refers mainly to figure 8d,e. "Contribution anomalies" meant contribution of AR forced rain to the >95$^{th}$ percentile precipitation and contribution AR forced rain to total precipitation.

**Line 488: RCA-IPSL and RCA-MPI have weaker responses than GFDL, CAN, and NORESM over the North Atlantic, don't they?**

Yes, we will change this accordingly.

**Lines 524-525: "This was done… period (Neiman et al. 2008)." I do not understand how this sentence justifies the use of a different 85th percentile in the historical and future experiments.**

We will rewrite this: The detection algorithm was developed based on weather data for the present-day reference period (1998-2005). During this reference period it was found that the 85$^{th}$ percentile  moisture content at noon (12:00) time steps corresponds to the median moisture contents found in ARs. Our figure 2 shows that the 85 percentile value strongly increases in the future climate. Hence, assuming that the relationship (85 percentile – median AR) is an intimate criteria to distinguish AR from the background we used the 85$^{th}$ percentile value calculated from the future climate.

**Lines 558-559: "Generally the AR imprint…eastern Europe." Isn't this sentence in contradiction with Fig. 5 where larger values are found over western Europe?**

Yes, in case of western France and the UK this is true. In case of Iberia the impact in ERAI is lower. We will adapt this sentence in a revised version.

**Technical issues:**

**Consider using commas much more often than currently.**

Thank you very much for the careful reading. We will adapt all the found typos and technical issues. We will also pass the manuscript a professional language service before any re-submission.

Line 22: "ER" -> ERA

Line 24: "eat" -> east

Lines 34, 426: "maximal" -> maximum

Line 37: "Iberia(15" -> Iberia (15

Line 40: "likely the originate" -> likely originate

Line 41: "from >60 ºN" -> from latitudes >60 ºN

Line 64: "Pacific Sectors of the World Ocean" -> Pacific sectors

Lines 75, 79, 194: "Laver" -> Lavers

Line 80: "However they" -> However, they

Line 90: "AR" -> ARs

Line 103: "framework employed" -> frameworks

Line 114: "4.5,RCP8.5" -> 4.5, RCP8.5

Line 118: "a validation RCA" -> a validation of the RCA

Line 135: "2005)and" -> 2005) and

Caption of Fig. 1: "Bathymetricy" -> Bathymetry

Line 165: "hindcasst" -> hindcast

Line 180: "W/m2" -> W/m$^2$

Line 206: Remove "Then".

Lines 223-224: "for each of the ensemble members respectively" -> for each model

Caption of Fig. 3: "below the threshold" -> below the 85th percentile

Caption of Table 3: "number ARs" -> number of ARs

Caption of Fig. 4: "b):" -> b)

Line 253: "AR" -> ARs, "Fig. 4" -> Fig. 4b

Line 281: "Bretagne" -> Brittany (to be consistent with line 579)

Line 300: Add parenthesis before Fig. 5.

Lines 279, 290, 299, 300, 422: "Figure" -> Fig.

Line 304: "respectively calculated as 0.98" -> respectively 0.98

Line 306: remove "respectively"

Line 314: "is" -> are

Caption of Fig. 6: "detercted" -> detected, "precentage" -> percentage, "AR" -> ARs

Line 323: "is" -> are

Line 325: "of spatial" -> of the spatial

Line 332: "ERAI-" -> ERAI

Line 333: "and larger" -> and a larger

Lines 335, 559: "distal" -> distant

Line 336: "This implies ARs" -> This implies that ARs

Line 338: "as in in semi aride": remove one "in"

Line 339: "effect the of" -> effect of

Line 346: "Fig. 5b" -> Fig. 5d

Line 360: remove "which is notably lower"

Line 383: "frequency" -> number

Line 398: "5b) no" -> 5b) but no

Line 400: "Figure 8c shows the number" -> Figure 8c shows that the number

Line 404: remove "Apart from this"

Line 412: "Figures" -> Figure

Line 420: "stronger" -> more

Line 421: "response" -> responses, "is" -> are

Line 424: the reference to the paper of Teichmann et al. 2018 is missing in the reference section.

Line 440: add comma between "latitudes" and "it" and remove comma after "found".

Line 445: "main driver AR" -> main driver of AR

Line 446: "Jet" -> jet

Line 451, 454: maybe not use "incidents" but rather events

Line 455: "for high" -> for the high

Line 460: remove "degree"

Lines 473-474: "sectors" -> sector

Line 483: "frequency of ARs" -> number of days with ARs

Line 484: "similar RCA" -> similar to the RCA

Line 489: "RCA-ECE show" -> RCA-ECE shows

Lines 490-491: "one realisation shows wider" -> RCA-MIROC shows wide, and remove "(RCA-MIROC)" at the end of the sentence.

Line 496: add comma after RCA-MPI

Line 501: "the most heavy" -> the heaviest

Line 504: "is" -> are

Lines 507-508: remove at least one of the "likewise".

Caption of Fig. 11: "forrcing" -> forcing

Line 519: "2016; 2016;" either a reference is missing or there is one "2016" too much.

Line 541: "Norway(Fig. 11c)" -> Norway (Fig. 11c)

Line 551: "model with" -> model ensemble with, "was applied" -> was created

Line 552: "regionalization" -> downscaling

Line 555: "the contribution to" -> the contribution of ARs to

Line 563: "climate" -> climates

Line 571: "of orographic" -> by orographic

Line 575: "show ARs" -> show that ARs

Line 588: "favors" -> favor

Line 589: "stronger" -> more strongly

Line 596: "arriving Scandinavia" -> arriving to Scandinavia

---

## Author Comment (AC4)

**In the present work, the authors used downscaled versions of global climate simulations from the CMIP5 ensemble using RCA-NEMO to study the atmospheric rivers over Europe in the present and future climates and compared the same with runs using ERA-I and using different RCPs. Results show that ARs will become more frequent and stronger in the future. Furthermore, the authors highlighted the variability of precipitation in different scenarios and different model simulations.**

**Though the study consists of some interesting results, the presentation of results, description and quality need to be improved. Information on the methods and quality of the figures including the methodology needs to be elaborated. Many typos, Grammar, missing words are found. Therefore, I would recommend the manuscript for major revision before accepting it for publication.**

We thank the reviewer for a thorough review of our manuscript and his/her valuable suggestions that really help to improve the manuscript. We regret that we were too short in the description of the methods and other parts of the text. We will be more comprehensive in a revised version and give lacking details.

**Major comments:**

**L100: Please elaborate on the purpose of the study.**

We will make the purpose of this study more clear. The main purposes are:

1) Presentation of the first AR analysis of downscaled CMIP5 model ensemble.
2) Investigate added value of high resolution for AR analysis
3.) Investigate future changes in AR characteristics

**L105: Please rephrase the sentence and carefully check the text corrections.**

We will split this sentence into two sentences and apply corrections.

**Section 2.1:**

**Please give more details on the model setup, spatial and temporal resolutions of climate models, comparison with ERA-I etc.**

We regret that some information is missing in the current manuscript. We will make clear this in another version.

**Also, it would be interesting to see a case study showing how later boundary forcing is influencing the AR characteristics in the RCA-climate model.**

The lateral boundary forcing is extracted from the global CMIP5 models and prescribed at the lateral boundaries. There is no feedback from RCA to outside the model domain. This means ARs enter the model domain at the lateral boundaries from the parent global model but then they develop freely and independent from the parent global model. We will make this more clear in a revised version.

A comparison of  ARs representation in RCA with ARs in the parent global model would be of course interesting but would require exhaustive additional analysis work for each of the global models. Therefore we think, such case study would be better pursued in a specific process study. For the conclusion of the current study we think this is not necessary.

**Section 2.2:**

**Table 2: Please provide a reason for leaving out the RCP2.6 cases for RCA – IPSL-CM5A-MR, RCA – CanESM2 and RCA – CNRM-CM5 despite their availability to force the RCA**.

There were technical reasons why these scenarios are lacking. We will add this information. The producer of the model ensemble, our co-author Christian Dieterich meanwhile suddenly deceased shortly after our submission of the current paper. Therefore we can not give more technical details here.

**Section 2.3: In addition to figure 2, a figure with the IVT threshold difference w.r.t historical mean in different RCPs**

**would give a robust picture of the difference in magnitude of IVT thresholds.**

[Figure]

The above plot shows the differences (future minus historical) of the IVT thresholds for the respective GHG scenarios. The plot gives indeed a more robust picture and allows more insights. For example, the yellow curve (RCA-HAD) suggests dynamical changes in addition to the thermodynamic effect of increased moisture load in a warmer climate. Hence, in RCP26 and RCP45 at 35°N the RCA-HAD IVT increases by 10% and 15%, but decreases in RCP85. This could be interpreted e.g. with a meridional shift of the core zone of westerly moisture transport in RCA-HAD.

**How do the discrepancies in IVT thresholds from different models and different RCPs are attributed to? Please give a brief note on the reasons for the biases in the IVT thresholds. Could it be due to bias in RCA runs or due to bias in the lateral boundary conditions from the GCMs?**

The discrepancies are most likely related to the different climates provided by the global models at the lateral boundary. First, all parent global models have a different thermodynamical equilibrium state and with this air temperatures and moisture load will be different. In connection with this, the large scale atmospheric circulation will differ (e.g. Brands, 2021) among the global models as the equator to pole temperature gradients are likewise different (Harvey et a., 2013). Consequently, the large differences in the IVT thresholds are not so surprising but he magnitude of difference is noteworthy. We will add a short paragraph to discuss these discrepancies.

Brands, S.: A circulation-based performance atlas of the CMIP5 and 6 models for regional climate studies in the northern hemisphere, Geosci. Model Dev. Discuss. [preprint], https://doi.org/10.5194/gmd-2020-418, in review, 2021

Harvey, B.J., Shaffrey, L.C. & Woollings, T.J. Equator-to-pole temperature differences and the extra-tropical storm track responses of the CMIP5 climate models. *Clim Dyn* **43,** 1171–1182 (2014). https://doi.org/10.1007/s00382-013-1883-9

**Figure 3: How do authors justify the large difference in IVT of ARs in hindcast and reanalysis? Though both reanalysis and hindcast data use observations, I would assume that the model parameters such as lead time, assimilation methods etc might be causing the bias here. Please discuss the same.**

[Figure]

We have to apologize for the bad quality of Figure 3 in the manuscript which makes it difficult to distinguish ERAI-RCA and ERA properly. We replotted Figure 3 and marked both curves with symbols (see above). Hence, notable differences between ERAI (black curve triangles) and ERAI-RCA (black curve filled squares) occur only south of 40°N. The largest difference, equals ~11% (at 35°N). That's not so much but nevertheless notable. Due to the rotated grid of RCA, the positions 35°N and 40 °N at 10°W are closest to the lateral boundary of RCA. Thus, the most likely explanation is, that we see here an effect of the errors associated with the lateral boundary coupling which are well known in regional models with one-way coupling to the driving coarse resolution models (e.g. Davies, 2013; Chikhar and Gauthier 2017).

Furthermore, the southern model boundary at 35°N lies in the transition zone from dry air masses of the subtropics and the wet air masses of the humid westerlies i.e. where high gradients in moisture content over short distances can be expected. Here, small differences in the mean position of the transition zone can cause large differences in local moisture content. We will discuss this in a potentially revised version.

Davies, T. (2014), Lateral boundary conditions for limited area models. Q.J.R. Meteorol. Soc., 140: 185-196. https://doi.org/10.1002/qj.2127
Chikhar, K., & Gauthier, P. (2017). Impact of Lateral Boundary Conditions on Regional Analyses, *Monthly Weather Review*, *145*(4), 1361-1379.

**Section 3.1:**

**L235: I would agree with the authors that the RCA impact on detecting the number would be less. However, it is to be noted that RCA could impact the strength of the ARs including the length to width ratio, footprint over a region, persistence etc. Please discuss these issues here.**

Good point. We agree RCA leaves it's own footprint on ARs within the domain and with this it influences geometry, duration and intensity. The alteration, and thus RCAs imprint on ARs, is likely stronger the more the RCA domain extends westward over the open Atlantic. We will include a short paragraph on this.

**Table 3: Why is RCA-MIROC showing a different trend in frequency expressed as the number ARs detected in a 30 year period of different climate scenarios? I suspect the value 445 8 from RCP45 in RCA-CAN. Please verify it.**

This was a mistake RCA-CAN has a value of 445. Thank you for the correction.
We agree there is apparently a different trend in RCA-MIROC. It is very likely that we see here dynamical effect that outcompedes the thermodynmic effect of increased temperature/ moisture load. This could indicate a shift in the meridional position of the westerlies in the global model or local circulation changes in RCA for example due to different warming patterns over land and over sea. We will include a remark in the manuscript to discuss this. However, this is difficult to diagnose in our regional model without further expensive experiments.

**Section 3.2:**

**L290: I would recommend authors include more details on rainfall variability by describing the dynamics and synoptic conditions responsible, rather than simply showing some stats.**

Our main point here was to show that even despite ARs are a very rare phenomenon (< ~3% of all days in a 30 year period are impacted by ARs) they can contribute significant amounts to the local yearly rainfall amount particularly in dry regions.

We agree more details on the dynamics of synoptic weather system would be interesting. However, this would need much more statistical analysis on detecting synoptic weather systems and would go beyond the purpose of this paper. However, there is literature about relationship between ARs and synoptical weather systems and we will include appropriate references (e.g. Pasquier et al., 2018).

**Section 3.2.3:**

**L325: Please verify the Figures numbers cited here. Through precipitation patterns are modulated by stochastic processes and further modulated by topography, the magnitude of IVT is affected by the small scale processes such as fluxes, SST and winds etc. Please go through the literature and discuss the same here.**

We meant Fig. 5c and 5d here (not 5b). Our main point here is that the high resolution RCA exhibit much more spatial noise here than the ERAI dataset. We attribute this to the different resolution of ERAI (0.75°) and RCA (0.22°). We appreciate the suggestion to discuss the relevant small scale processes responsible for that in a potentially revised version.

**L335: Please describe the reasons for the bias in RCA ARs inland propagation.**

This is likely related to a more rapid moisture loss of AR over land in RCA. It could also be related to the fact that in RCA-NEMO the North Sea and Baltic Sea is interactively simulated with a an high resolution ocean model, while in the ECMWF-IFS model that produced the ERAI reanalysis these two seas are implemented only a fixed surface scheme. This means basically that SST, sea ice boundary fields are prescribed from ERA data while in RCA4-NEMO they are calculated during simulation. We will describe this in more detail and include a short paragraph on this.

**Section 4.2:**

**L385: Not sure if the figures cited here are relevant to the context. Please verify**
we will carefully check this and change if necessary.

**L400: Increase in AR forced heavy precipitation over Eastern Europe may also associate with changes in absolute path of the ARs, increase in IVT/moisture availability and duration/persistency of the ARs over the land. Please discuss these issues.**

Good point! We completely agree. We will definitely add a paragraph on this.

**Section 4.3:**

**It is a good practice to present figures with lat-long labels which are missing for almost all spatial figures in the manuscript. Please redraw them in the revised manuscript.**

We agree and will redraw all the maps.

**L450: The approach selected by the authors in finding the source region of ARs raises many questions. For example, earlier the authors mentioned that As originate from open oceans. But taking 10W as a reference for finding the source region does not line up with the earlier statement and may induce errors in results. Furthermore, it is not customary to find the source region according to the AR incidence/landfalling.**

Sorry, we have to pass the manuscript to a professional native speaker before potentially resubmitting. We did not want to evaluate the source region (unfortunately we used this wording) but we meant the meridional bin where ARs were for the first time was detected. We agree this doesn't say anything about the source region where the ARs formed. As you mentioned in your previous comment (L400), this says something about the absolute path of the AR after entering Europe at 10°W. We will rewrite the whole section and make the purpose of this section more clear.

**L460, 465, 470: Please rephrase the sentences with proper citations to the figures.**
Will be done.

**L475: " we can conclude that ARs from the southern Atlantic sectors are more present over most land regions in a warmer climate" is a strong statement in this context.**
We will rephrase this and make it consistent with our analysis

**Section 5.1: Please describe how and why the higher latitudes experience increasing precipitation despite the decrease in ARs.**

We will make this more clear in an appropriate place in 5.1. ARs do also increase in the high latitudes >°60°N but ARs <45°N increase even more. Consequently the relative contribution on land from ARs entering Europe north of 60°N lowers. The more "southern ARs" may carry warmer and wetter air masses.

**P495: How the decreased AR impact over Norway can be explained with the decrease of ARs arising from >60N. Earlier authors claim that Scandinavian ARs originate from the south.**

Yes, we claim here that in the future climate, ARs originate more from latitude <60° thereby carrying warmer air and more moisture. We will rephrase this paragraph and make this more consistent with other text passages.

**L515: A figure showing regional mean precipitation change in the historical/future scenarios from individual models with error bars would give a better idea of the magnitude of precipitation changes.**

That's true. However, in the context here we aim to explicitly refer to the strengthened AR influence on heavy precipitation. The mean changes even on a seasonal resolution have been investigated in Gröger et al. (2021, https://link.springer.com/article/10.1007/s00382-020-05489-8, Figs 2 & 3).

**Section 6:**

**L590: Figure 9 shows the opposite result. Please explain the same.**
We agree, we will remove this sentence.

**Minor comments:**

**1. Please carefully go through the text and sentences and correct the typos, values, and grammar.**

**2. Authors are requested to re-check the references as some of them are not matching the context they are cited.**

**3. Please improve the quality of the figures**

We agree the manuscript needs a careful major revision of figures and and in particular language style.

---

## Author Response (AR1)

Reply to review RC1

**This study discusses the ARs over the North Atlantic from ensembles of 24 global climate simulations following the greenhouse gas scenarios RCP2.6, RCP4.5, RCP8.5 downscaled using a regional climate model (RCA-NEMO) with 0.22° resolution and the results are compared with against ER-I reanalysis data. The study finds that ARs would become more frequent and more intense in a future warmer climate especially in the higher emission scenarios under the assumption of RCP2.6. They also propagate further inland to eastern Europe in a warmer climate.**

**Though I am yet to complete the review, here is a major comment on the detection of atmospheric rivers. Authors have mentioned that they employed Lavers et al. (2012, 2013) method to detect ARs based on the 5 degrees binning along 10 degrees west. Though it is a well-known approach, one might see the spatial "patchy" and "noise" at a given time step in the AR detection (figure 1 in Lavers et al., 2013). To be specific, one might expect that a high-resolution data detection algorithm could retain values over a few grid points that satisfy the binned threshold but do not satisfy the AR criteria. This noise in turn would cause bias in comparing the long term (climatology etc) spatial variability among different models. Also, the authors mentioned that the ARs with 18 hours of persistence were considered. But I do not see any description of finding persistence. Hence, authors are encouraged to provide more details on these issues in the manuscript.**

We thank the reviewer for this important note. It's difficult to properly extract this kind of noise which would require to develop advanced methods of filtering etc. This would be out of the scope of the present manuscript. However, we agree that this should be mentioned in the section "3.4 Effect of downscaling". We included the following paragraph at the end section 3.3.2 (line 422):

"Finally, we note that the comparison between different spatial resolutions might also reflect different noise levels. This noise occurs when isolated grid points located outside the AR at a given time step exceed the IVT threshold but do not satisfy the geometric and temporal requirements. The different noise levels would then contribute to the total effect of downscaling."

We regret the confusion with the description of the algorithm. In the new version the algorithm is now described step by step in section "2.3 Detection of atmospheric rivers".

With persistence we mean "duration", i.e. when the algorithm detects an AR for less than 18 hours (corresponding to 3 consecutive output time steps), then these time steps are not retained and the atmospheric river is no longer classified as such (following basically Lavers et al (2012, 2013). We have made clear this in the revised version (step 7. in section 2.3).

Reply to reviewer RC2

**This paper investigates the behavior of North Atlantic ARs in reanalysis and ensembles of global simulations in historical and future simulations. The manuscript is in good shape. It is well-written and has a clear structure. I don't have major concerns for the paper, but some minor suggestions and comments for discussion:**

We thank the reviewer for a thorough review of our manuscript and his/her valuable suggestions that really helped to improve the manuscript.

**L193: Guan and Waliser (2015) does not provide a thorough review of AR detection methods, instead, GW15 explained their AR detection algorithm in detail. Since GW15's algorithm is not adopted in this paper, it is better to cite a different paper as the "overview". For example:**

**Shields et al. (2018): Atmospheric River Tracking Method Intercomparison Project (ARTMIP): project goals and experimental design, Geosci. Model Dev., 11, 2455-2474, https://doi.org/10.5194/gmd-11-2455-2018, 2018.**

Thank you for this correction. We have replaced Guan and Waliser (2015) by Shields et al., 2018 in the revised version.

**Figure 4: I wonder how the duration distribution would change if the duration threshold to tuned to 6 hours or 12 hours. How will the duration (including those excluded short-lived events (<18hrs)) change in the future climate?**

We have have recalculated this exemplary for the MPI-ESM realization. The main effect is, as expected, an increase of about 55% in the total of detected ARs. The left hand figure below compares the distributions calculated with a threshold set to 18 hours (orange) and 12 hours (blue). Generally, both distributions are similar and indicate most ARs in the short term fraction (with >50% ARs lasting 30 hours or shorter). The right hand figure displays the relative change for the RCP8.5 scenario [%, i.e. future minus historical period *100 / historical period)]. In both distributions the tendency towards extra long (48 hours or more) ARs can be seen. However, these extended ARs remain typically rare (<10%) in both historical and future climate.

[Figure]

**Figure 5a: instead of showing the actual number of AR days, it might be helpful to show the percentage of AR days to the number of days of the period.**

We fully agree. In the revised version we have adopted Figure 5a accordingly. Likewise Figure 8a which displays the change in AR day frequency in a future warmer climate is adopted in the same way.

**Table 3 and L367: Is it possible that the dynamical field is more active in the future simulation as well?**

**Table 3 shows that the number of AR increased in the future runs: it makes sense if the historical AR threshold is applied in the future run – however, here the future ARs are detected with IVT thresholds calculated from future simulation, so the higher moisture load is reflected in the future IVT threshold to some extent. Therefore, I am curious if the change in dynamical filed also contribute to the increase in future AR number.**

Yes, it is likely that both dynamical as well the thermodynamical changes contribute to the increase in AR number. Hence, it could be that more ARs are initiated in the source region (outside our model domain) or that the pathway from the source area to Europe changed. This can not be analyzed in our regional model. However, this question was addressed in global CMIP5 models (which are more suited for this question). Lavers et al. (2013) argue for mostly thermodynamical changes (moisture increase) for the future increase in ARs. Gao et al., (2016), likewise found thermodynamics playing a dominant role but discussed also potential dynamical changes ARs.

Lavers, D. A., Allan, R.P., Villarini, G., Lloyd-Hughes, B., Brayshaw, D.J., Wade, A.J.,, Future changes in atmospheric rivers and their implications for winter flooding in Britain, *Environ. Res. Lett.* **8,** 034010, DOI: 10.1088/1748-9326/8/3/034010, 2013b

Gao, Y., Lu, J., & Leung, L. R., Uncertainties in Projecting Future Changes in Atmospheric Rivers and Their Impacts on Heavy Precipitation over Europe, Journal of Climate, 29(18), 6711-6726; DOI: https://doi.org/10.1175/JCLI-D-16-0088.1, 2016.

**L450: I am not sure if I fully understand the methodology. Are the origins of ARs being categorized by the southernmost latitude? If so, isn't it more related to the curvature of AR's shape than the actual origin?**

It's not the southernmost. First, for every AR that is detected over land we check the time and the latitudinal position where the AR strikes the 10°W meridian. For this we used AR masks generated during the detection. For the post-statistical analysis we then summarize by regions south of 45° degree between 45-60° and >60° to obtain the results. The method is now fully described step by step in section 2.3 "Detection of atmospheric rivers".

**L563: please rewrite, grammar problem.**

Done. The paragraph:

"The regional climate model is further used to investigate the dynamics of ARs in present and future climate. For this in an ensemble of in total 34 simulations was carried to downscale global climate model scenarios from the CMIP5 suite with a coarse resolution between 1.4 – 3°. The models were used to downscale three greenhouse gas scenarios (RCP2.6, RCP4.5, and RCP8.5)."

reads now:

"The regional climate model was further used to investigate AR in present and future climates. Thus, an ensemble of global CMIP5 climate simulations (1.4°–3° resolution) was downscaled to reach a 0.22° resolution. In total, 34 simulations were carried out for the greenhouse gas emission scenarios RCP2.6, RCP4.5, and RCP8.5. "

Reply to reviewer RC3

**The authors make use of downscaled versions of global climate simulations from the CMIP5 ensemble to study the atmospheric rivers reaching Europe in the present and future climates. They find that ARs will become more frequent and stronger in the future, especially in the RCP8.5 scenario. The results also show that the orientation of ARs will change with for example more ARs coming from the south reaching Scandinavia (Norway).**

**The study is interesting but I got confused with some of the fields presented in the figures, which quality is not great. Some information about the methods and the fields displayed in the figures is missing, the methodology and results are not well discussed and compared to previous studies, and there are many technical mistakes (typos, English, missing words). Therefore, I think the manuscript needs major revision in order to be in a publishable state.**

We thank the reviewer for a thorough review of our manuscript and his/her valuable suggestions that really help to improve the manuscript. We regret that we were too short in the description of the methods and other parts of the text.

We have now profoundly revised the manuscript considering recommendations off all reviewers. Major changes include:

1. A "step by step" description of the AR detection method is included in section"2.3 Detection of atmospheric rivers"

2. New section "2.4 Detection in future climate" explains how AR are detected in the future runs. The reason why we used IVT 85[th] percentiles from the future climate are described and the consequences for the interpretation are given.

3. New section "2.5 Calculation of indices" describes precisely how climate indices in Figures 5, 8, and 12 have been derived. Figures 5 and 8 are now fully consistent (showing the same set of indices).

4. In section "5.1 Uncertainties with respect to the choice of CMIP5 models" We added a paragraph that shows that uncertainty (i.e. model spread) is generally higher in the extreme precipitation regime than in the low or mean precipitation regime.

5. Section "4.3 Influence of dynamical changes" is completely rewritten. The method has been made clear now and the interpretation is more concise.

6. All figures have been redrawn. They now contain visible lon/lat labels. Maps do show the same domain.

Please find our point to point replies below:

**Major comments:**

**About the methods:**

**- The temporal scale of the RCA model outputs is not clear. From line 209, it seems to be 6-hourly but it would be great to also mention it in section 2.1 or in section 2.2. In the same line, is the extreme precipitation determined from 6-hourly or daily data?**

All data analyzed for this study has 6-hourly resolution. In the new version we noted this at several places. We included this as own column in Table 1 (section 2.1). Also, it is additionally noted in the new description of the detection algorithm (2.1), and in the new section "2.5 Calculation of indices". It is also noted and in the new paragraph decscribing the ERAI reanalysis data at the end of chapter "2.2 The high resolution ensemble".

**- The authors use an AR minimum length threshold of 1500 km. This is quite short compared to previous studies and I guess the reason for this low value is the limited extension of the domain to the west. Can the authors state the reason for such threshold and maybe discuss it in light of previous studies?**

In this study we followed the previous approach from Lavers and Villarini (2013) who used a minimum length of 1500 km to detect ARs in an ERA-I reanalysis product. However, we agree that due to our limited domain the algorithm does not detect

1. ARs that do not reach Europe but remain outside over the Atlantic Ocean. Thus, our study can not be compared 100% to global CMIP studies on ARs that take into account also those ARs.

2. Over the western Iberian Peninsula, which is located relatively close to the western model boundary, some AR could be missed or detected with delay (as it may take longer to reach the 1500 km criterion). Over the UK and Norway this does not play a significant role as these countries lie far away from the models lateral boundary.

These aspects are now included in the method section 2.3 (Detection of atmospheric rivers) under step 5.

Lavers, D. A., and Villarini, G. , The nexus between atmospheric rivers and extreme precipitation across Europe, *Geophys. Res. Lett.*, 40, 3259– 3264, doi:10.1002/grl.50636, 2013

**- Can the authors confirm that only one AR is detected at every timestep? What happens when an AR covers two latitudinal bins and/or exceeds the IVT 85th percentile in two adjacent 5º bins?**

Yes, we can clearly confirm this, as this is checked by our algorithm. ARs are detected separately for each of the 5° latitudinal bins. At this stage it is indeed possible that one AR time step is recorded twice (in two litutudinal bins). Therefore, after detection, the whole AR record is checked for double time steps (year,  month, day, and hours information is saved). All double time steps are removed.

This is now explained in section 2.3 Detection of atmospheric rivers under step 6.

**- Do the authors define a mask of every AR to link them to precipitation?**

Exactly, for every single AR a mask array is created which contains likewise information about the exact date and time. All this in 6-hourly resolution. This is mentioned now in section 2.3 under step 8.

**- Lines 223-224: I think the authors should emphasize the fact that the AR detection threshold is different between the present and future periods in contrast to previous studies and what the advantage of this method is. As for now, it is written later in the manuscript (section 5.2) but it should appear upfront. I believe that this choice limits the influence of the larger moisture content in the atmosphere in the future climate on the results. Is that true? Can this aspect be discussed if relevant?**

We also think this fact should be emphasized more prominently already in the methods section.

As stated in Lavers et al., (2012; 2013) the $85^{th}$ percentile of all 12:00 (noon) time steps represents approximately the median value of moisture content within real observed ARs in todays climate (1998-2005). Thus, our approach  conserves the relationship between the median moisture content of ARs with the $85^{th}$ percentile of all noon time steps. The other way to do would be to apply the historical threshold also to the future atmosphere which contains much more moisture also in the background field. This emphasizes more the thermodynamical aspect but does not ensure full compliance with the algorithm developed for present day AR characteristics.

We have put this description and the discussion in the new section "2.4 Detection in future climate".

**- I believe the ERA-Interim reanalysis is not really described. For example, the spatial and temporal resolution used in the study is missing.**

We now added a short paragraph summarizing information on ERAI relevant to our study (last paragraph section 2.2, line 184). Additionally, Table 1 contains most important parameters for the ERAI data set and compares it to the data sets derived from the RCA climate model.

**- In the caption of Fig. 2 and line 329, it is written that ARs are tracked. However, the AR tracking is not explained. I suspect the tracking is used to check the AR persistence and involves ARs masks. Therefore, please add this information in section 2.3.**

Yes, the word tracking is misleading here (as it suggests a kind of Lagrangian approach). We did not really track ARs  but, as you said, checked persistence of ARs over time. This has been corrected (section 2.3, step 7)

**About Section 3.2.3:**

**Lines 351-354: Can the authors explain a bit more how a negative temperature bias in the regional climate model is linked to a "too high moisture load" and to the higher number of ARs in September in the hindcast compared to ERA-Interim? I would expect a higher temperature to be linked to more moisture. Moreover, wouldn't it be more useful to look at the precipitation in the hindcast, in ERA-Interim, and in the E-OBS dataset, instead of looking at the temperature? Could it also be useful to assess the difference in the specific humidity between the hindcast and ERA-Interim?**

We have strongly revised section 3.2.3. Effect of Downscaling" (now section 3.4 in the new version). This section focuses now on the comparison between the hindcast RCA-ERAI and ERAI to show the added value of high resolution. Therefore we have removed the paragraph on a potential temperature bias because this was too speculative.

**Lines 354-356: This sentence must be rewritten. It is not clear at all.**

The sentence is now removed (see previous comment above).

**What would be the impact of using the E-OBS precipitation instead of ERA-Interim's in the results displayed in the right column of Fig. 5b-d?**

We recalculated figures 5b-d (below) using the AR masks from ERA-I combined with precipitation from the E-OBS data set (rightmost column). We note, that by doing so, we violate the IVT equation shown in line 203 which means there is no physical consistency between the atmospheric moisture content and precipitation. However, the result is shown in the right-most column. For the percentage of yearly maximum precipitation attributed to ARs (%AMP, see below figure 5b), the spatial pattern is more or less the same. Differences occur over Norway and the UK where ERAI_eobs_rain shows a weaker signal compared to ERAI.

With respect to the fractional contribution of AR induced precipitation to the total >95th percentile precipitation (Figure 5c) and the fractional contribution of AR induced precipitation to the total amount of precipitation (Figure 5d) we note that the pattern is the same as for ERAI (third column) but the amount of rain related to ARs is overall higher compared to ERAI.

[Figure]

*Revised Fig. 5. a) AR frequency expressed as total number of days a grid cell was covered by an AR during the historical period. b) Percentage of annual maximum precipitation related to ARs (%AMP) c) fractional contribution of AR forced precipitation to the >95th percentile precipitation (%95P). d) same as c) but for the total precipitation (%TP).(Note that in a) the unit has been changed from "total number AR days within 30 years" to %AR days of total days as recommended by reviewer RC2).*

**About section 5.2:**

**Gao et al. (2016) showed that ARs became more frequent north because of the poleward shift of the eddy-driven jet. That is not what the authors seem to obtain by distinguishing the "origin" of the ARs. I believe it would be more interesting and useful to look at the eddy-driven jet response in all scenarios to explain the responses in the number of days with ARs and AR-forced precipitation?**

That's true. Gao et al. found in CMIP5 models a relationship between the low level jet position in historical climate and future changes in ARs .

We agree it would be interesting to look on the eddy driven jet in our RCA simulations. However, due to its limited domain RCA excludes significant part of the North Atlantic ocean. Therefore it is not suited to analyze changes in the position of the Jet Stream as this feasible in global models. Please also note our revised section "4.3 Influence of dynamical changes". In

the previous version we gave the impression of a shift in the ARs source region. That was wrongly formulated. Indeed, we mean changes in the route ARs take east of 10°W and across Europe. Therefore, our results are not contrary to Gao et al. (2016) but complementary. We made this clear in last paragraph of section 5.2 Differences to global projections (lines 623ff).

**Remove sentences "At least… (Ma et al. 2020). The authors…likely reasons." on lines 546-548 because it is not relevant for the present discussion.**

We agree and removed the two sentences in the revised version.

**About the figures' content:**

**- Figure 2: is the 85th percentile determined using all points within the 5º bins and all time steps? In any case, please state somewhere how the values shown in this figure were calculated.**

Yes, it was calculated from all the grid cells in the range of the respective 5° bin and all time steps. It's now stated in section "2.3 Detection of atmospheric rivers"  under steps 2 and 3.

**- Figure 4, 7: how is the "average moisture transport over land" calculated? Are the authors using a mask for the ARs and averaging the IVT within the mask? Please explain.**

Yes, this is exactly the way it was done. It is now explained in section 2.3 under step 8.

**- Figure 5: Row a): the authors describe this row as AR frequency in the caption but, as written in the text, it is only a number of days. One would have to divide by the total number of days in the period to get a frequency. Moreover, since the model output is 6-hourly, I do not understand how the authors convert it to a number of days. What if an AR covers two days during its lifetime? Is it counted twice? Also, if the AR lasts a minimum of three time steps (because of the 18h minimum duration) in the same day, it is counted only once, correct?**

For the analysis we classified a day as AR day if within the 24 hours at least one AR event was recognized. That means, even if only one of the four 6 hour time steps of day was impacted by an AR, this day is counted as AR day. Consequently, any AR that covers two subsequent days (lasting from 18:00 to 12:00 of the following day) is counted as 2 days. The explanation is now included in the new section 2.5 Calculation of indices.

In the new version Figs 5a, and 8a AR days are now calculated as frequency (AR days divided by total number of day in 30 years) and expressed as %.

**Rows b), c), and d): I find not clear what is shown in those panels. It is worth mentioning in the text how those "indices" are calculated and keep the same names throughout the manuscript. Can the authors explain why ARs contribute to the yearly maximum (row b) over southwestern Norway or northern UK but barely contribute to the extreme precipitation (row c)?**

We regret the confusion that our description of the indices caused. See our exact explanation below.

**Row b): I understand this figure as the percentage of years (among the 30 years of the period) for which the maximum precipitation occurs when there is an AR. Is that correct? In any case, the text should be clarified. The same comment applies to the rows c) and d).**

Exactly, if in 15 years, out of the 30 year period, the annual maximum precipitation can be related to an AR we get a value of 50%.

In figure c) we

1. sum up the accumulated precipitation that occurs in all rain events that exceed the 95th percentile rain rate.

2.  sum up the accumulated precipitation that occurs in all rain events that exceed the 95th percentile rain rate and can be related to ARs

3. we calculated how big is the fraction (%) of the sum in step 2 to the sum calculated in step 1.

In figure d)

we do same. But instead considering precipitation events >95 th percentile only we consider all rain events (not only the >95th percentile events).

In the new version we now denote this "fractional contribution of ARs to total precipitation (%TP) or to the >95th percentile precipitation" (%95P). This is now described in the new section "2.5 Calculation of indices".

**Row c): how do the authors relate the low values for the Norwegian coast to the study of Benedict et al. (2019) who found that 85% of the extreme precipitation events are related to ARs?**

**Benedict, I., K. Ødemark, T. Nipen, and R. Moore (2019): Large-Scale Flow Patterns Associated with Extreme Precipitation and Atmospheric Rivers over Norway. Mon. Wea. Rev., 147, 1415-1428.** https://doi.org/10.1175/MWR-D-18-0362.1

If we understand Benedict et al., (2019) right, then they calculated the percentage of the number (N) of >99th percentile rain events. This is not what we did. As done in Gao et al. (2016), we calculated the accumulated volume of precipitation (not the number of events; see above). In the below figure we applied the approach of Benedict et al. (2016) to our RCA-ERAI run. As can be seen, over southwestern Norway, up to 80 % of >99 percentile events are related to ARs which is in well agreement with Benedict et al., (2019).

[Figure]

*% of number of AR related >99th percentile rain events to total number of >99th percentile rain events (as done in Benedict et al., 2019).*

**- Figure 8: is this figure showing a simple difference between the future and the historical experiments or does it show a relative change?**

Figure 8a (AR days) shows a simple difference; not the relative change. Likewise, for figures 8b, 8c, 8d) and 8e) simple differences were shown (i.e. the % values for the future minus the % values for the historical). We have clarified this in the caption of Fig. 8 in the revised version.

**Caption: I would not call what is displayed "climatological indices". Please remove. Panel b) does not show "precipitation rates" if it is similar to Fig. 5. Please use the same wording for Fig. 8 as for Fig. 5.**

We agree and have fully harmonized figures 5 and 8 in the new version. All indices are described now in section "2.5 Calculation of indices"

**"Note all non-robust" -> Note that all non-robust**

We changed this in the Fig. 8 caption in the new version.

**What is the difference between panels c and d? One of the two is not shown in Fig. 5.**

Figure 8c was showing the change in % of number of events in the 95th percentile (similar to Benedict et al., 2016 but for the 95th percentile instead of the >99th percentile fraction). We have removed 8c as it added not significant information to what was shown in the other panels. Figs. 5 and 8 display now the same set of indices.

**Why is the unit in panel c is % if the what is shown is "the number of of events" as written on lines 400-401?**

Panel 8c is now removed from Fig. 8 (see our answer to the above comment).

**- Figure 10: Are the authors sure that panel a) is for ARs originating north of 60ºN and panel b) for ARs originating south of 45ºN? It does not seem in agreement with the text. For example, the sentence "the RCA ensemble clearly shows a relative increase of those ARs originating south of 45ºN (Fig. 10a)." However, Fig. 10a only shows negative values and the caption says that panel a) is for ARs originating north of 60ºN. Please make sure that the caption and text (from line 458 to 475) correspond to the figure. Can the category 45-60ºN be displayed as well? In the caption, what does "relative contribution" mean? Does it show the relative difference between the future and historical experiments or is it a simple difference?**

This was a mistake. It should be

"the RCA ensemble clearly shows a relative increase of those ARs originating south of 45ºN **(Fig. 10b)**." (Figure 10b in the previous version showed increase of originating ARs south of 45°.)

In the new version we have also included the category 45-60°N. The section "4.3 Influence of dynamical changes", which discusses Fig. 10, has been revised. The main conclusion is that AR from the south increase stronger than ARs from the north. This increases their relative share over land nearly everywhere.

**- Figure 11: Why does the color bar for the standard deviation panels exhibit negative values? A standard deviation is positive. Is the STD panel a difference between the future and historical standard deviation or the standard deviation of the responses displayed?**

The standard deviation denotes the inter-model spread of the responses. We have now completely redrawn Fig. 11 (Figure 12 in the revised version) which is now in the same style as the other 2D maps. No negative values in the scale of standard deviation any more.

**About the figures' quality:**

**I think the quality of the figures should be improved.**

**Figures 2, 3, 4, 8, 9, 10, and 11 exhibit gray frames around the panels and around the color bars. Could they be removed?**

The frames were introduced when they were inserted in our word processing package. The original png figure files have no visible frames. We will deliver the original png files to the Copernicus office and so we hope there will be no frames in the published version.

**Figure 9 exhibits a colored line in between the two panels as if another figure was below.**

The figure was accidentally inserted twice. This has been fixed.

**Figures 1, 5, 9, and 10 exhibit weird coastlines over Greece. Moreover, Crete and the Balearic Islands are missing.**

We ave tried to make the plots with higher resolved coastlines. However, it turned out, that at higher resolution the regions with "noisy coastlines" come out as black opaque areas due to multiple lines plotted at short distance. In these regions (e.g. coasts of UK, France Norway) this completely covered the color shades and so destroyed the important information. Therefore, we decided to keep the less coarser resolved coastline for the benefit of the information we want to show.

**Figure 11 has too small labels for the latitudes (longitudes missing) and for the color bars. Moreover, the coastlines are discontinued at 20ºE.**

This figure has now been redrawn and now contains larger lon/lat information. It is figure 12 in the new version.

**Figures 4, 6, and 7: the frames, tick marks, and background grids are almost invisible. Please make them darker or black.**

Grids, frames and tick marks have been made thicker.

**Figures 7, and 11, and Tables 2 and 3: Please arrange the GCMs in alphabetical order as the authors did for Fig. 2.**

**It would be great if all figures showed the same domain.**

The figure/tables are now rearranged in alphabetical order. All maps show the same domain now.

**Figure 8: can the columns be rearranged such that RCP2.6 is on the left and RCP8.5 on the right? I find it more intuitive and it would be consistent with Figs. 2 and 7 and Tables 2 and 3.**

The columns are now rearranged accordingly.

**Figure 2: the gray and yellow lines are barely visible.**

Figure 2 has been revised now. Grey and yellow lines turned to dark green and orange respectively.

**Figure 5: In the left column, panels b), c), and d) should have RCA_MEAN as title instead of RCA_ENSM. Better remove the titles of the three bottom rows as it would make the panels bigger, improving their readability.**

We now used consistently RCA_MEAN instead of RCA_ENSM. Also, we move the titles inside the maps which allowed larger maps.

**Minor comments:**

**Lines 25-28: it seems from this sentence that Norway is in Central Europe. Please rewrite this sentence, maybe splitting it in two.**

We agree and removed the second part of the sentence. Itread now:

"Over central Europe, the models simulated a smaller propagation distance of AR toward eastern Europe than obtained using the ERAI data."

**Sometimes, the authors write ERA-I and sometimes ERAI. Please be consistent in the text and captions and make sure the same acronym is used everywhere (I suggest ERAI).**

We now use consistently ERAI.

**Line 138: why is the coupling between the RCA model and NEMO only over the North and Baltic seas? What about the Mediterranean and the Norwegian seas?**

The regional setup of NEMO was originally developed for the North Sea and Baltic Sea which requires a very high spatial resolution (~3.7 km on average, and 56 vertical levels). Hence NEMO needs extremely high computational resources. Therefore the other sea had to be excluded. This is now explained in section "2.1 The regional climate model RCA"

**Line 151: "in a huge ensemble": to which ensemble do the authors refer to? Does it have a name? This ensemble seems similar to the one used in the present study so why not using it?**

Yes it's the same ensemble. We have changed the sentence now (line 159):

" It has been employed in previous studies to investigate the present climate and simulate the mean response to global climate change by downscaling global climate scenarios (Dieterich et al., 20119….)"

**Caption of Figure 1: Please rewrite it. This figure mainly shows the topography of the domain in brown and the bathymetry in blue-green colors.**

We have now corrected the caption Fig. 1.

**The authors very often use very "greenhouse gas scenarios". I suggest to use instead "greenhouse gas emission scenarios" or probably better "GHG emission scenarios".**

We now use consistently "greenhouse gas emission scenarios" throughout the manuscript

**Lines 202-203: please add the units of g, the wind, and dp.**

Units of g [$m^3/(kg * s^2)$)] Wind [m/s] and *dp* [Pa] are now added.

**Line 203: "In the two hydrostatic models…": I assume that the authors here refer to the regional climate model RCA and to the IFS model. Why does the reader need this information?**

We agree and removed this sentence.

**Caption of Fig. 2 : "IVT thresholds" -> 85th percentile of IVT**

**"…for the ensembles' historical…" -> … for all models and the historical…**

We changed the caption accordingly

**Line 278 (page 10): "heavy precipitation": please write that this is defined using the 95th percentile in order to relate it to what is shown in Fig. 5c.**

We agree this is important. In the new version we have included a complete description of all indices in section "2.5 Calculation of indices"

**Lines 279-280: What are "the mean climatic conditions in Europe"? Moreover, it rains a lot in Southwestern Norway and that is not reflected in Fig. 5a.**

This mainly refers to the different climates found in Europe, i.e. wetter conditions in Northern Europe / dryer conditions in southern Europe.

What we basically want to show is: in very humid regions with plenty of rain like Norway, ARs can not contribute so much to the total and to the >95$^{th}$ percentile rain as they do in dryer regions (like Iberia). We have rewritten the sentence:

"The spatial pattern mainly mirrored the AR frequency pattern but it also reflected the varying long-term mean hydrological conditions in Europe:…" (line 381)

**Line 296: What is meant with "weather regimes"? Moreover, what is the point of lines 295-298?**

We have reformulated this section and removed these lines.

**Line 321: Do you mean effect of the downscaling? With "regionalization", it sounds to me like the authors split the domain into different regions.**

We used the two term synonymous. We now replaced "regionalization" by "downscaling" to avoid any confusion.

**Line 324: Can the authors explain the "factor of 10"?**

The factor refers to the grid cell size in RCA (550-600 $km^2$) and the ERAI reanalysis (~6000 $km^2$) given in table 1 of the former manuscript. We have changed this now and noted the grid distances in x and y metrics (~24*24km for RCA; 80*45 km for ERAI).

**Line 378: "AR" -> ARs**

Changed.

**The reference Massoud et al. (2020) does not seem appropriate here as the paper is not about the US but about the Middle East. Maybe the authors meant the following reference:**

**Massoud, E. C., H., Lee, P. B., Gibson, P., Loikith, and D. E., Waliser (2020): Bayesian Model Averaging of Climate Model Projections Constrained by Precipitation Observations over the Contiguous United States, Journal of Hydrometeorology, 21, 2401-2418. https://doi.org/10.1175/JHM-D-19-0258.1**

Thank you for this correction. The reference has been changed.

**Lines 418-419: what is meant with "contribution anomalies"? Could it be replaced by responses?**

This has been rephrased to %95P and %TP and is now consistent with the new description of indices in the methods section (section 2.5 Calulation oof indices)

**Line 488: RCA-IPSL and RCA-MPI have weaker responses than GFDL, CAN, and NORESM over the North Atlantic, don't they?**

Sorry, we apparently confused the individual plots. Thank you for the correction. We have replotted the figure 12 and revised the text. Also the maps were reordered alphabetically.

**Lines 524-525: "This was done… period (Neiman et al. 2008)." I do not understand how this sentence justifies the use of a different 85th percentile in the historical and future experiments.**

We have put the explanation now in the newly added section "2.4 Detection in future climate" in the methods section where also the consequences are discussed.

**Lines 558-559: "Generally the AR imprint…eastern Europe." Isn't this sentence in contradiction with Fig. 5 where larger values are found over western Europe?**

Yes, in case of western France and the UK this is true. In case of Iberia the impact in ERAI is lower. We have changed the sentence accordingly.

Over Iberia, the AR imprint on the analyzed indices in the ERAI data set was weaker but stronger in distant parts of eastern Europe.

**Technical issues:**

**Consider using commas much more often than currently.**

Thank you very much for the careful reading. After the revision of the text and the figures the manuscript was by a native speaker.

**Line 22: "ER" -> ERA**

done.

**Line 24: "eat" -> east**

done.

**Lines 34, 426: "maximal" -> maximum**

changed consistently throughout the manuscript

**Line 37: "Iberia(15" -> Iberia (15**

done.

**Line 40: "likely the originate" -> likely originate**

done.

**Line 41: "from >60 ºN" -> from latitudes >60 ºN**

done.

**Line 64: "Pacific Sectors of the World Ocean" -> Pacific sectors**

done

**Lines 75, 79, 194: "Laver" -> Lavers**

corrected.

**Line 80: "However they" -> However, they**

done.

**Line 90: "AR" -> ARs**

done.

**Line 103: "framework employed" -> frameworks**

done.

**Line 114: "4.5,RCP8.5" -> 4.5, RCP8.5**

done.

**Line 118: "a validation RCA" -> a validation of the RCA**

done.

**Line 135: "2005)and" -> 2005) and**

done.

**Caption of Fig. 1: "Bathymetricy" -> Bathymetry**

done.

**Line 165: "hindcasst" -> hindcast**

corrected.

**Line 180: "W/m2" -> W/m$^2$**

done.

**Line 206: Remove "Then".**

done.

**Lines 223-224: "for each of the ensemble members respectively" -> for each model**

done.

**Caption of Fig. 3: "below the threshold" -> below the 85th percentile**

done.

**Caption of Table 3: "number ARs" -> number of ARs**

done.

**Caption of Fig. 4: "b):" -> b)**

done.

**Line 253: "AR" -> ARs, "Fig. 4" -> Fig. 4B**

done.

**Line 281: "Bretagne" -> Brittany (to be consistent with line 579)**

done.

**Line 300: Add parenthesis before Fig. 5.**

done.

**Lines 279, 290, 299, 300, 422: "Figure" -> Fig**.

Whenever referenced in parenthesis we have now consistently written "Fig." instead of "Figure". Only when it is at the beginning of a sentence we wrote Figure.

**Line 304: "respectively calculated as 0.98" -> respectively 0.98**

done.

**Line 306: remove "respectively"**

done.

**Line 314: "is" -> are**

this sentence has been rephrased to improve readabilty.

**Caption of Fig. 6: "detercted" -> detected, "precentage" -> percentage, "AR" -> ARs**

done.

**Line 323: "is" -> are**

**Line 325: "of spatial" -> of the spatial**

**Line 332: "ERAI-" -> ERAI**

**Line 333: "and larger" -> and a larger**

**Lines 335, 559: "distal" -> distant**

**Line 336: "This implies ARs" -> This implies that ARs**

**Line 338: "as in in semi aride": remove one "in"**

**Line 339: "effect the of" -> effect of**

**Line 346: "Fig. 5b" -> Fig. 5d**

**Line 360: remove "which is notably lower"**

Section "Effect of Downscaling has been completely rephrased (line 320-360, old bersion). Typos have been fixed.

**Line 383: "frequency" -> number**

As stated above we have now recalculated the frequencies of AR-days in Figs 5a and 8a.

**Line 398: "5b) no" -> 5b) but no**

the sentence has been rephrased.

**Line 400: "Figure 8c shows the number" -> Figure 8c shows that the number**

the sentence has been rephrased.

**Line 404: remove "Apart from this"**

the sentence has been rephrased.

**Line 412: "Figures" -> Figure**

done.

**Line 420: "stronger" -> more**

done.

**Line 421: "response" -> responses, "is" -> are**

done.

**Line 424: the reference to the paper of Teichmann et al. 2018 is missing in the reference section.**

Thank you. The reference is now added.

**Line 440: add comma between "latitudes" and "it" and remove comma after "found".**

**Line 445: "main driver AR" -> main driver of AR**

**Line 446: "Jet" -> jet**

**Line 451, 454: maybe not use "incidents" but rather events**

**Line 455: "for high" -> for the high**

**Line 460: remove "degree"**

**Lines 473-474: "sectors" -> sector**

The section lines 440 to 476 (old version) has been revised and typos have been fixed.

**Line 483: "frequency of ARs" -> number of days with ARs**

Figures 5a and 8a display now frequencies  (see above)

**Line 484: "similar RCA" -> similar to the RCA**

done.

**Line 489: "RCA-ECE show" -> RCA-ECE shows**

done.

**Lines 490-491: "one realisation shows wider" -> RCA-MIROC shows wide, and remove "(RCA-MIROC)" at the end of the sentence.**

Done.

**Line 496: add comma after RCA-MPI**

done.

**Line 501: "the most heavy" -> the heaviest**

done.

**Line 504: "is" -> are**

done.

**Lines 507-508: remove at least one of the "likewise".**

Removed "likewise" in line 507.

**Caption of Fig. 11: "forrcing" -> forcing**

done.

**Line 519: "2016; 2016;" either a reference is missing or there is one "2016" too much.**

This has been fixed

**Line 541: "Norway(Fig. 11c)" -> Norway (Fig. 11c)**

done.

**Line 551: "model with" -> model ensemble with, "was applied" -> was created**

done.

**Line 552: "regionalization" -> downscaling**

done.

**Line 555: "the contribution to" -> the contribution of ARs to**

done.

**Line 563: "climate" -> climates**

done.

**Line 571: "of orographic" -> by orographic**

done.

**Line 575: "show ARs" -> show that ARs**

done.

**Line 588: "favors" -> favor**

done.

**Line 589: "stronger" -> more strongly**

done.

**Line 596: "arriving Scandinavia" -> arriving to Scandinavia**

done.

Reply to reviewer RC4

**In the present work, the authors used downscaled versions of global climate simulations from the CMIP5 ensemble using RCA-NEMO to study the atmospheric rivers over Europe in the present and future climates and compared the same with runs using ERA-I and using different RCPs. Results show that ARs will become more frequent and stronger in the future. Furthermore, the authors highlighted the variability of precipitation in different scenarios and different model simulations.**

**Though the study consists of some interesting results, the presentation of results, description and quality need to be improved. Information on the methods and quality of the figures including the methodology needs to be elaborated. Many typos, Grammar, missing words are found. Therefore, I would recommend the manuscript for major revision before accepting it for publication.**

We thank the reviewer for a thorough review of our manuscript and his/her valuable suggestions that really help to improve the manuscript. We regret that we were too short in the description of the methods and other parts of the text.

We have now profoundly revised the manuscript considering recommendations off all reviewers. Major changes include:

1. A "step by step" description of the AR detection method is included in section"2.3 Detection of atmospheric rivers"

2. New section "2.4 Detection in future climate" explains how ARs are detected in the future runs. The reason why we used IVT 85th percentiles from the future climate are described and the consequences for the interpretation are given.

3. New section "2.5 Calculation of indices" describes precisely how climate indices in Figures 5, 8, and 12 have been derived. Figures 5 and 8 are now fully consistent (showing the same set of indices).

4. In section "5.1 Uncertainties with respect to the choice of CMIP5 models" We added a paragraph that shows that uncertainty (i.e. model spread) is generally higher in the extreme precipitation regime than in the low or mean precipitation regime.

5. Section "4.3 Influence of dynamical changes" is completely rewritten. The method has been made clear now and the interpretation is more concise.

6. All figures have been redrawn. They now contain visible lon/lat labels. Maps do show the same domain.

Please find our point to point replies below:

**Major comments:**

**L100: Please elaborate on the purpose of the study.**

The main purposes of this study are to:

- conduct the first analysis of AR over Europe using a downscaled CMIP5 model ensemble
- Investigate the added value of high resolution in representing AR in a climate model.
- assess future climate related changes in AR characteristics over Europe.
- explore uncertainties with respect to the choice of the global model and in regard to the choice of the greenhouse gas emission scenario

We have included the text to make the purpose more clear.

**L105: Please rephrase the sentence and carefully check the text corrections.**

We have completely rephrased the paragraph. The whole text was corrected by a native speaker after revision.

**Section 2.1:**

**Please give more details on the model setup, spatial and temporal resolutions of climate models, comparison with**

**ERA-I etc.**

Table 1 includes now an overview about horizontal resolution (in km & degree), vertical levels, and temporal resolution for the climate model RCA and the used ERAI reanalysis data. We have also added a short paragraph that briefly summarizes the information about the ERAI reanalysis (ERAI) and ERAI hindcast simulation (RCA-ERAI).(last paragraph in section "2.2 The high resolution ensemble")

**Also, it would be interesting to see a case study showing how later boundary forcing is influencing the AR characteristics in the RCA-climate model.**

The lateral boundary forcing is extracted from global CMIP5 models/ERAI reanalysis and prescribed at the lateral boundaries. There is no feedback from RCA to outside the model domain. This means ARs enter the model domain at the lateral boundaries from the parent global model but then they develop freely and independent from the parent global model. We have added this information in the first paragraph (lines 141ff) at the end of section 2.1 to make this clear.

**Section 2.2:**

**Table 2: Please provide a reason for leaving out the RCP2.6 cases for RCA – IPSL-CM5A-MR, RCA – CanESM2 and RCA – CNRM-CM5 despite their availability to force the RCA.**

There were technical reasons why these scenarios are lacking (not all driving data necessary for RCA were available from these 3 models). We now added this information in the caption of table 2

**Section 2.3: In addition to figure 2, a figure with the IVT threshold difference w.r.t historical mean in different RCPs would give a robust picture of the difference in magnitude of IVT thresholds.**

[Figure]

*Figure 2b (revised version): change [%] of IVT thresholds in the future scenarios relative to the respective historical period.*

We have added the difference plots as Figure 2b in the revised version (see above).
The plot gives indeed a more robust picture and demonstrates the presence of dynamical changes. We added a short paragraph at the end of the new section "3.1 Differences in IVT thresholds in historical, and hindcast simulations":

"There was no evidence of a linear relationship between climate warming and the and the increase in IVT thresholds (Fig. 2b). For example, in RCA-HAD the IVT at 35°N in the low and moderate warming scenarios (RCP2.6 and RCP4.5) increased by ~10% and ~15%, respectively (Fig. 2b). In the strongest warming scenario (RCP8.5), however, the IVT decreased by ~5%. This suggests that, over the long term, dynamic changes influence the IVT.

**How do the discrepancies in IVT thresholds from different models and different RCPs are attributed to? Please give a brief note on the reasons for the biases in the IVT thresholds. Could it be due to bias in RCA runs or due to bias in**

**the lateral boundary conditions from the GCMs?**

The discrepancies are most likely related to the different climates provided by the global models at the lateral boundary. First, all parent global models have a different thermodynamical equilibrium state and with this air temperatures and moisture load will be different. In connection with this, the large scale atmospheric circulation will differ (e.g. Brands, 2021) among the global models as the equator to pole temperature gradients are likewise different (Harvey et a., 2013). Consequently, the large differences in the IVT thresholds are not so surprising but he magnitude of difference is noteworthy. We will add a short paragraph to discuss these discrepancies.

Brands, S.: A circulation-based performance atlas of the CMIP5 and 6 models for regional climate studies in the northern hemisphere, Geosci. Model Dev. Discuss. [preprint], https://doi.org/10.5194/gmd-2020-418, in review, 2021

Harvey, B.J., Shaffrey, L.C. & Woollings, T.J. Equator-to-pole temperature differences and the extra-tropical storm track responses of the CMIP5 climate models. *Clim Dyn* **43,** 1171–1182 (2014). https://doi.org/10.1007/s00382-013-1883-9

This is now discussed in the new section 3.1 Differences in IVT thresholds in historical, and hindcast simulations.

**Figure 3: How do authors justify the large difference in IVT of ARs in hindcast and reanalysis? Though both reanalysis and hindcast data use observations, I would assume that the model parameters such as lead time, assimilation methods etc might be causing the bias here. Please discuss the same.**

[Figure]

We have to apologize for the bad quality of Figure  in the old manuscript version which made it difficult to properly distinguish RCA-ERAI and ERAI. We replotted Figure 2 and marked both curves with symbols (see above). Hence, notable differences between ERAI (black curve triangles) and ERAI-RCA (black curve filled squares) occur only south of 40°N. The largest difference, equals ~11% (at 35°N). That's not so much but nevertheless notable. Due to the rotated grid of RCA, the positions 35°N  and 40 °N at 10°W are closest to the lateral boundary of RCA. Thus, the most likely explanation is, that we see here an effect of the errors associated with the lateral boundary coupling which are well known in regional models with one-way coupling  to the driving coarse resolution models (e.g. Davies, 2013; Chikhar and Gauthier 2017).

Furthermore, the southern model boundary at 35°N lies in the transition zone from dry air masses of the subtropics and the wet air masses of the humid westerlies i.e. where high gradients in moisture content over short distances can be expected. Here, small differences in the mean position of the transition zone can cause large differences in local moisture content. We will discuss this in a potentially revised version.

An explanation has been added to the new section "3.1 Differences in IVT thresholds in historical, and hindcast simulations."

Davies, T. (2014), Lateral boundary conditions for limited area models. Q.J.R. Meteorol. Soc., 140: 185-196. https://doi.org/10.1002/qj.2127

Chikhar, K., & Gauthier, P. (2017). Impact of Lateral Boundary Conditions on Regional Analyses, *Monthly Weather Review*, *145*(4), 1361-1379.

**Section 3.1:**

**L235: I would agree with the authors that the RCA impact on detecting the number would be less. However, it is to be noted that RCA could impact the strength of the ARs including the length to width ratio, footprint over a region, persistence etc. Please discuss these issues here.**

Good point. We agree RCA leaves it's own footprint on ARs within the domain and with this it influences geometry, duration and intensity.

We have added this point in the first paragraph of section "3.2 General statistics":

"...This indicates that the number of AR in the RCA is primarily controlled by the parent global model at the lateral boundaries. This is not surprising since AR develop in open ocean regions far outside the model's domain. However, within that domain the RCA develops freely, leaving its own fingerprint on AR, by controlling their intensity, geometry, and lifetime. The RCA fingerprint is likely to become stronger with growing distance from the lateral boundaries…."

**Table 3: Why is RCA-MIROC showing a different trend in frequency expressed as the number ARs detected in a 30 year period of different climate scenarios? I suspect the value 445 8 from RCP45 in RCA-CAN. Please verify it.**

This was a mistake RCA-CAN has a value of 445. Thank you for the correction.

RCA-MIROC shows indeed different trends. The reason for that is difficult to analyze and would need further analysis on the global model MIROC5 which is beyond the scope of the present study. We added the following remark in section "3.2 General statistics"

"Compared to the historical period, RCA-MIROC increased by 15% in RCP 4.5 but only by 5% in the strongest warming scenario (RCP8.5, Table 3). Hence, there was no linear scaling with global mean warming. This can be explained by the changes in the large-scale circulation in the parent global model, such as induced by shifts in the eddy-driven jet (Gao et al., 2016)."

**Section 3.2:**

**L290: I would recommend authors include more details on rainfall variability by describing the dynamics and synoptic conditions responsible, rather than simply showing some stats.**

Our main point here was to demonstrate that even despite ARs are a very rare phenomenon (< ~3% of all days in a 30 year period are impacted by ARs) they can contribute significant amounts to the local yearly rainfall amount particularly in dry regions. We have rephrased section 3.2 (3.3 un the new version) to make this more clear.

We agree that more details on the dynamics of synoptic weather system would be interesting. However, this would need much more statistical analysis and would require to develop algorithms for the detection and classification of synoptic weather systems in the RCA model output. This would go beyond the purpose of this paper. However, there is literature about relationship between ARs and synoptical weather systems (Pasquier et al., 2018). We made a reference to the Pasquier et al paper at the end of section 5.2.

Pasquier, J. T., Pfahl, S., & Grams, C. M. (2019). Modulation of atmospheric river occurrence and associated precipitation extremes in the North Atlantic Region by European weather regimes. Geophysical Research Letters, 46, 1014– 1023. https://doi.org/10.1029/2018GL081194

**Section 3.2.3:**

**L325: Please verify the Figures numbers cited here. Through precipitation patterns are modulated by stochastic processes and further modulated by topography, the magnitude of IVT is affected by the small scale processes such**

**as fluxes, SST and winds etc. Please go through the literature and discuss the same here.**

We have strongly revised this section (section "3.4 Effect of downscaling" in the new version) which now mainly focuses on the effect of resolution. Therefore, in this context we would like to avoid a review on small scale processes like winds surface temperatures and air temperatures etc. that influence the magnitude of the IVT. Instead, we added a reference to Feser et al. (2011) Hoheneger et al. (2020) and Stevens et al. (2020) and Gomez-Navarro et al. (2011) that discuss the added value of high resolution on convection processes and the representation of topography in high resolution models. (Line 399ff).

**L335: Please describe the reasons for the bias in RCA ARs inland propagation.**

This is difficult to explore. It could be related to different model physics between RCA and the ECMWF IFS model (which produced the ERAI reanalysis with data assimilation). It could also be related to the fact that in RCA the North Sea and Baltic Sea is interactively simulated with a high resolution ocean model, while in the ECMWF-IFS model these two seas are implemented only as a fixed surface scheme. This means basically that SST, sea ice boundary fields are prescribed from ERA data while in RCA4-NEMO they are calculated during simulation.

We now discussed possible reasons in an own paragraph of section "3.4 Effect of Downscaling" (lines 411ff).

**Section 4.2:**

**L385: Not sure if the figures cited here are relevant to the context. Please verify**
Section 4.2 "Spatial changes" was substantially revised and Fig. 8C in the old version was removed. This was done to harmonize Figs 5 an 8 which now show the same parameters. We rephrased the text body and likewise checked references to figure as well.

**L400: Increase in AR forced heavy precipitation over Eastern Europe may also associate with changes in absolute path of the ARs, increase in IVT/moisture availability and duration/persistency of the ARs over the land. Please discuss these issues.**

This comment refers to Fig. 8c of the old version which showed high relative increases (%) of AR events over eastern Europe. We have removed this Figure 8c to harmonize Figures 8 and 5 to show the same parameters (as recommended by other reviewers). The aspect of changed absolute AR paths / moisture availability is discussed now in section "4.3 Influence of dynamical changes"

**Section 4.3:**

**It is a good practice to present figures with lat-long labels which are missing for almost all spatial figures in the manuscript. Please redraw them in the revised manuscript.**

We have redrawn all the maps and added longitude/latitude labels.

**L450: The approach selected by the authors in finding the source region of ARs raises many questions. For example, earlier the authors mentioned that As originate from open oceans. But taking 10W as a reference for finding the source region does not line up with the earlier statement and may induce errors in results. Furthermore, it is not customary to find the source region according to the AR incidence/landfalling.**

Sorry, this approach was described insufficiently in the previous version. We did not want to evaluate the source region (unfortunately we used this wording) but we meant the latitudinal bin where ARs strike the 10°W meridian. We agree this doesn't say anything about the source region where the ARs formed (outside our model domain). As you mentioned in your previous comment (L400), this says something about the absolute path of the AR after entering Europe at 10°W.

In the revised version, the Section "4.3 Influence of dynamical changes" has been substantially revised to more accurately describe the analysis method, it's interpretation and the conclusions.

**L460, 465, 470: Please rephrase the sentences with proper citations to the figures.**
We have rephrased the two paragraphs and corrected some mistakes in the reference to figures.

**L475: " we can conclude that ARs from the southern Atlantic sectors are more present over most land regions in a warmer climate" is a strong statement in this context.**
This statement is misleading. We removed it in the new version.

**Section 5.1: Please describe how and why the higher latitudes experience increasing precipitation despite the decrease in ARs.**

This was poorly described. ARs do also increase in the high latitudes >60°N but ARs <45°N increase even more. Consequently the relative contribution on land from ARs entering Europe north of 60°N lowers.

We described this now more clearly in the revised section "4.3 Influence of dynamical changes". We also added a note in section "6 Summary and conclusions" (line 663).

**P495: How the decreased AR impact over Norway can be explained with the decrease of ARs arising from >60N. Earlier authors claim that Scandinavian ARs originate from the south.**

In the future climate AR increase at all latitudes, but the relative contribution from <45° increases while the relative contribution from >60°N decreases. This is described now in the revised section "4.3 Influence of dynamical changes"

**L515: A figure showing regional mean precipitation change in the historical/future scenarios from individual models with error bars would give a better idea of the magnitude of precipitation changes.**

That's true. However, in the context here we aim to explicitly emphasize the strengthened AR influence on heavy precipitation. The mean changes even on a seasonal resolution have been investigated in Gröger et al. (2021, https://link.springer.com/article/10.1007/s00382-020-05489-8, Figs 2 & 3).

**Section 6:**

**L590: Figure 9 shows the opposite result. Please explain the same.**

This sentence is removed in the new version.

**Minor comments:**

**1. Please carefully go through the text and sentences and correct the typos, values, and grammar.**

**2. Authors are requested to re-check the references as some of them are not matching the context they are cited.**

**3. Please improve the quality of the figures**

We have strongly revised the manuscript and the text was revised by a native speaker. Also the figures quality has been improved.

---

## Referee Report (RR1)

Atmospheric Rivers in CMIP5 climate ensembles downscaled with a high-resolution regional climate model.

Review report:

Many thanks to the authors for addressing the review comments carefully and revising the manuscript accordingly. I strongly believe that the manuscript has been improved a lot through changes and additions made by the authors. Despite the changes made, I would recommend authors to make **a minor revision** to improve the article further and make it suitable for publication in the journal Earth System Dynamics.

Comments:

1. Ln-23: Please write full form or expand ERAI at first instance.
2. Authors may discuss more about interpolation techniques used in comparing downscaled RCA-ERAI with ERAI reanalysis data.
3. Ln-25: Please improve the abstract by providing reasons for the results discussed in the abstract.
4. Ln-42: Authors may discuss these results in connection with the North Atlantic winter storm activity and frequency changes.
5. Ln-78:  Is it local groundwater recharge?
6. Ln-136: I would assume that authors used HIRLAM based Rossby Centre regional atmospheric climate model in which atmospheric model RCA and ocean model NEMO were used. However, in the text, it is mentioned that the Rossby Centre regional atmosphere model, which might be confusing to the reader. Please make necessary corrections carefully.
7. Ln-186: Please detail the interpolation techniques if any are used.
8. Ln-225: Though it is a well-known approach for Europe, I suspect a drawback back in it. Here authors are comparing the 85th percentile along 10 degrees west with westward and eastward grid cells. This approach might suites well for the westward grid cells where in many instances, moisture transport occurs from the west or southwest. But assuming the same threshold for the eastward grid cells might eliminate a few grid cells from the actual AR imprint. This is because the AR landfall at the western Europe boundary could decrease IVT along the eastern landmass due to the loss of IVT in the form of precipitation and the moisture cut-off from the ocean. Hence the less IVT values during the subsequent time steps in a given AR. As pointed out in line 366, Instead, authors may consider taking the 85th percentile at each grid specially and find the AR.
9. Ln-274: Should it be RCP 8.5? Please check Figures 7 and 8 labels as well.
10. Fig-3: It would be nice to see an additional figure displaying a special difference between reanalysis and hindcast.
11. Ln-285: It is not entirely clear whether authors consider AR days with precipitation or without precipitation in computing ARF.
12. Ln-327: Please check and correct the sentence.
13. Ln-330: Please try to elaborate on the dynamics influencing the IVT decrease in RCA-HAD in the RCP 8.5. For example, you may consider citing the recent paper from Venugopal et al., (2021), who have studied the decadal changes in specific humidity and wind components leading to changes in IVT over the North Atlantic using reanalysis data.
14. Ln-438: Is it RCA-MEAN in place of RCA-ENSM?

15. Ln-445: Very interested to see that RCA-MEAN failed to reproduce the precipitation but IVT. Please consider including more details and causes for the same for the reader's benefit.
16. Ln-547: The statement is valid if the North Atlantic SSTs are strong enough to permit the convection in summer. But in winter, the scenario mostly depends on the Arctic cold air outbreaks and convergence of the moisture from the adjacent areas into the AR.
17. Fig-12: Does the inter-model standard deviation means the average standard deviation obtained from different RCA ensembles?
18. Ln-635: Please check the sentence as it is contradicting.

---

## Author Response (AR2)

**Response to review report 1**

**I acknowledge the great effort made by the authors in improving their manuscript and responding to my comments. The manuscript was much nicer to read and clearer but I still have some minor comments listed below following the manuscript order.**

We thank the reviewer for a thorough second round review. We carefully considered all recommendations which really helped to further improve the manuscript.

**Abstract, Lines 19-20: I think the authors should add here the downscaling of 9 historical simulations and of the ERAI reanalysis as well.**
We agree and have complemented the information about the hindcast and historical imulations

**Table 2:**
**- The time period 2070-2099 should be written too for the three RCP.**
Done.

**- In the caption, I find "the original ERAI hindcast data" a bit misleading. Could the authors replace it with the ERAI reanalysis?**
Done.

**Section 2.2, Line 184: The title is also misleading because the authors write about the hindcast produced with the ERAI reanalysis. I suggest to remove this small title as it is not needed.**
We agree and removed the title.

**Section 3.3:**
**- Line 206: the unit of the gravitational acceleration should be [m s^-2] and not "[m^3/(kg * s^2)]".**
Of course. This corrected now.

**- The authors could also mention here the unit of the IVT as well (kg m^-1 s^-1).**
We agree and have included the unit in line 200.

**- Point 8 from line 245 onwards: The authors should mention if they use an absolute value or a percentile of IVT as threshold to define the AR mask. From Fig. 3, the mask seems to be defined with IVT above the 85th percentile of each grid point.**
The masks were generated by mapping the exceedance of the percentile thresholds shown in Figure 2. We have made clear this now.

**Section 2.5, Line 304: I believe the word "heavy" should not appear here.**
We removed the word "heavy" here.

**Section 3.2:**
**- Lines 338-339: Maybe refer to Fig. 3 at the end of the sentence "However, … and lifetime.".**
Good idea. We included the reference to Fig. 3.

**- Line 350: after "warming", I would add "compared to the other models".**
Done.

**- In the paragraph made by lines 348-351: since the authors emphasise the different behaviour of MIROC, they could maybe also add that RCA-MIROC is the model with the lowest number of AR detected in all simulations (historical and future), as far as I can see in Table 3.**
We now noted that the increase for RCA-MIROC is lowest among the models in RCP8.5 (line 355).

**Table 3: some white spaces are missing between the numbers and parenthesis in column 3 for RCP2.6.**
Thank you. We have inserted the lacking white spaces now.

**Section 3.3, Line 367: The reference to Fig. 2 is not needed here and also a bit confusing.**
We removed the reference.

**Section 3.4, Lines 421-425: do the authors mean that there might not be only one IVT mask for some time steps?**
No. Any double entries for IVT masks were removed during the detection (as stated section 2.3, page 7, point 6)

**Section 3.5:**
**- Lines 438-439: "ENSM" -> MEAN. I believe the references to the figures are wrong. It should be: 0.86 for %AMP (Fig. 5b), 0.82 for %TP (Fig 5d), and 0.92 for %95P (Fig. 5c). I suggest to switch the last two to get the figures in alphabetical order (b, c, and d).**
We turned ENSM → MEAN.
Thank you for the correction. The figure references are now in the right order.

**- Line 451: Between the two sentences "…hindcast." and "However, …", I would also point out that RCA-MEAN and RCA-ERAI highly overestimate the number of AR in September compared to ERAI but underestimate it in October. This was mentioned in the previous version of the manuscript (see lines 748-760 of the tracked-changes version). The authors do not need to provide an explanation for this behaviour.**
We re-included the following sentence at that place:
"In addition, RCA-MEAN and RCA-ERAI highly overestimate the number of AR in September compared to ERAI but underestimate it in October."

**Figure 7: "Note no realization are available for RCA-IPSL, RCA-CAN and RCA-CNRM." Please add "that" between "Note" and "no" and mention that it is for rcp26 only at the end.**
Done.

**Figure 8: "Difference between 2070-2099 minus 1970-1999" -> Difference between 2070-2099 and 1970-1999. The caption should also mention that the figure displays the ensemble mean difference.**
Fully agree. We added this important information in the caption now.

**Section 4.2:**
**- Line 502: "nearly everywhere." -> nearly everywhere for RCP45 and RCP85.**
**- Line 511: "The changes" -> The relative changes.**
Thank you for the corrections. They are included now.

**Figures 9 and 10: both figures have a different projection than the rest of the maps (Figs. 5, 8, and 12).**
Yes, but since the differences in the projections are of cosmetic nature and so we think this is acceptable.

**Figure 9: as for Fig. 8 mention in the caption that this is an ensemble mean of the relative change.**
Done.
**Figure 10: same comment as for Figs. 8 and 9: this is an ensemble mean.**
Done.

**Section 4.3, Line 538: Yes, there is a reduction over Norway but also over many European mountainous regions, such as the Pyrenees, Massif Central, Alps, mountains in the Balkans. The authors could also mention the high absolute values In Scandinavia in panels (a) and (c) of Fig. 10 which seem to be linked to the lack of sea ice in the Gulf of Bothnia in the future.**
We agree and now mentioned also the other mountainous regions. We are a bit careful pointing explicitly to a potential relationship to the sea ice retreat in the Gulf of Bothnia without further analysis. However, this definitely merits investigation in a follow up process study.

**Section 5.1:**
**- Line 556: What do the authors mean with "models' solutions"? Should it be responses instead of "solutions"?**
We replaced "model solutions" by "model results".
**- Line 566: I would remove "highly" because there is some variation among the models.**
Done.

**- Line 659: "Fig. 8a" -> Fig. 12a.**
Thank you for the correction!

**- Line 575: "along the Norwegian coast": This statement can be extended to the whole Scandinavia.**

Done.

**- Line 580: "smaller" than when? I believe it should say smaller in the future (compared to the historical period).**
Correct. We added "in the future" at this place.

**- Line 591: add RCA-ECE here as well.**
Done.

**- Line 592: "decrease, occurring in the UK; was" -> decrease over UK was.**
Done.

**Figure 12: please add in the caption that the figure is valid for scenario RCP85.**
Done.

**Section 5.2:**
**- Line 617: "a robust climate change signal": I would not call it "robust" but high.**
We now changed the sentence to " a strong climate signal"

**- Lines 625-627: instead of or in addition to the low-level jet stream, I would mention the storm tracks with a reference to Zappa et al. (2013) their Fig. 2a, which I think is going in the direction of the present study's results.**
We agree this is more specific and fits better in the context. We included the reference to Zappa et al. (2013) and storm tracks.

**Section 6:**
**- Line 635: "distinct N-S gradients": to me, the Iberian peninsula rather shows E-W gradients in the ERAI reanalysis. RCA-ERAI shows gradients in both directions because of the orography.**
Fully correct. We rephrased this sentence accordingly (line 648).

**- Lines 636-637: This sentence should be rewritten as it is not clear. Suggestion: The AR imprint on the analysed indices in the ERAI dataset was weak over Iberia but stronger in distant parts of Eastern Europe compared to the downscaled RCA-ERAI (Fig. 5).**
Thank you for the suggestions. We changed the text accordingly.
**- Line 678: "is coarse" -> is still relatively coarse.**
Done.

**Technical details:**

**Line 79: "(Albano et al., 2017; )" Is there a reference missing? Otherwise, should be (Albano et al., 2017).**
Changed to (Albano et al., 2017).

**Line 83: "Held and Soden ,2006" -> Held and Soden, 2006**
Done.

**Lines 112,113,114: put commas at the end of the three lines.**
Done.

**Line 220: "bin specific" -> bin-specific**
Done.

**Lines 230 and 231: remove the commas in the two references (after Villarini and after al.).**
Done.

**Line 322: "equator to pole" -> "equator-to-pole"**
Done.

**Line 327: delete one "and the".**
Done.

**Line 390: "Downscling" -> Downscaling**
Done.

**Line 442: "occurs" -> occur**
Done.

**Line 538: "a~20%" -> a ~20%**
Done.
**Line 539: "Fig.10a" -> Fig. 10A**
Done.

**Line 588: "standard deviations were" -> standard deviation was. The authors write about only one standard deviation here, the one displayed in Fig. 12c.**
Done.

**Line 627: "Pasquier et a., 2018" -> Pasquier et al., 2018**

**Response to review report 2**

**Many thanks to the authors for addressing the review comments carefully and revising the manuscript accordingly. I strongly believe that the manuscript has been improved a lot through changes and additions made by the authors. Despite the changes made, I would recommend authors to make a minor revision to improve the article further and make it suitable for publication in the journal Earth System Dynamics.**

We thank the reviewer for a thorough second round review. We carefully considered all recommendations which really helped to further improve the manuscript.

**Comments:**
**1. Ln-23: Please write full form or expand ERAI at first instance.**
We now wrote the full form at line 23.

**2. Authors may discuss more about interpolation techniques used in comparing downscaled RCA-ERAI with ERAI reanalysis data.**
We have added a note in line 188 that we use the bilinear remapping technique provided by "climate data operators" package (Schulzweida et al. 2021) where more details can be found. However, a robust discussion about interpolation techniques would require more analysis which is not the scope of the present paper.

Schulzweida, U.: CDO User Guide (Version 2.0.0). Zenodo. http://doi.org/10.5281/zenodo.5614769, 2021.

**3. Ln-25: Please improve the abstract by providing reasons for the results discussed in the abstract.**

We added the discussed changed storm track activity over the North Atlantic and/or changed weather regimes as potential driver of the found local changes in the AR pathway over Europe (last sentence of the abstract; see also point 4 below).

However, we would like to avoid to make the abstract too long and too detailed but rather provide a summary of the main results. A condensed text summarizing main results and the underlying reasons is already provided in section 6 "Summary and conclusion" at the end of the article.

**4. Ln-42: Authors may discuss these results in connection with the North Atlantic winter storm activity and frequency changes.**
Good idea. We now added the sentence here:
"The found changes in the local AR pathway are probably driven by larger scale circulation changes such as a change in dominating weather regimes and/or changes in the winter storm track over the North Atlantic"

**5. Ln-78: Is it local groundwater recharge?**
Basically yes, in particular in semi-arid regions. We mentioned this explicitly in the next sentence (line 81).

**6. Ln-136: I would assume that authors used HIRLAM based Rossby Centre regional atmospheric climate model in which atmospheric model RCA and ocean model NEMO were used. However, in the text, it is mentioned that the Rossby Centre regional atmosphere model, which might be confusing to the reader. Please make necessary corrections carefully.**

We slightly adapted the sentence to be more precise:

"The atmospheric part of the regional climate model (Wang et al., 2015; Gröger et al., 2015; Dieterich et al., 2019) is based on Rossby Center regional atmosphere model RCA (Samuelsson et al., 2011; Kupiainen et al., 2014) version 4."

The RCA model version 4 was then coupled to the NEMO3.3 high resolution model (North Sea + Baltic Sea) as introduced by Wang et al. (2015) and Dieterich et al. (2019).

**7. Ln-186: Please detail the interpolation techniques if any are used.**

We used the bilinear remapping technique. This is now mentioned in the last paragraph of section 2.2 " the high resolution ensemble (line 190).

**8. Ln-225: Though it is a well-known approach for Europe, I suspect a drawback back in it. Here authors are comparing the 85th percentile along 10 degrees west with westward and eastward grid cells. This approach might suites well for the westward grid cells where in many instances, moisture transport occurs from the west or southwest. But assuming the same threshold for the eastward grid cells might eliminate a few grid cells from the actual AR imprint. This is because the AR landfall at the western Europe boundary could decrease IVT along the eastern landmass due to the loss of IVT in the form of precipitation and the moisture cut-off from the ocean. Hence the less IVT values during the subsequent time steps in a given AR. As pointed out in line 366, Instead, authors may consider taking the 85th percentile at each grid specially and find the AR.**

We agree the standard approach may be subject to some inaccuracies especially over distal land land parts when the AR becomes weaker. However, we are hesitant with the proposed solution, i.e. taking the local 85 percentile, which we think would more robust empirical evidence and tests against the standard method. This would require much more research and is definitely beyond the scope of the present paper. Therefore, we here decided to rely on the established approach as the other studies did for Europe.

**9. Ln-274: Should it be RCP 8.5? Please check Figures 7 and 8 labels as well.**

Thank you. We corrected the typo. We also harmonized the format of the labels in Figures 7 and 8 with the labels in the text and in the other figures.

**10. Fig-3: It would be nice to see an additional figure displaying a special difference between reanalysis and hindcast.**

We principally agree. However, the results discussed in the text are already clear and well supported by comparing the middle column (RCA-ERAI) and the right hand column in Figure 5. Therefore, we think an RCA-ERAI - ERAI difference plot is not necessary here in the context of the text.

**11. Ln-285: It is not entirely clear whether authors consider AR days with precipitation or without precipitation in computing ARF.**

We did not distinguish this. The analysis of ARF is based on IVT only. We added the sentence

"No distinction was made whether an AR day was associated by precipitation or not."

at line 293 to make this clear.

**12. Ln-327: Please check and correct the sentence.**

Done.  A double "and the" was removed from the sentence (see also the above response to report 1 (line 327)).

**13. Ln-330: Please try to elaborate on the dynamics influencing the IVT decrease in RCA-HAD in the RCP 8.5. For example, you may consider citing the recent paper from Venugopal et al., (2021), who have studied the decadal changes in specific humidity and wind components leading to changes in IVT over the North Atlantic using reanalysis data.**

Thank you for the reference. We have added a note on the potential role of decadal variation in wind components and specific humidity with a reference to the study of Venugopal Thandlam (line 337):

"Also decadal variations in wind components  and specific humidity have to be considered in this context (Thandlam et al., 2021)."

**14. Ln-438: Is it RCA-MEAN in place of RCA-ENSM?**

RCA-ENSM has been changed to RCA-MEAN now.

**15. Ln-445: Very interested to see that RCA-MEAN failed to reproduce the precipitation but IVT. Please consider including more details and causes for the same for the reader's benefit.**

We can only speculate on that and can not give a final explanation based on our results. However, we added the following note (line 454):
"We note that the exact reproduction of precipitation pattern is difficult in coarse resolution models due to insufficient description in cloud physics and/or the treatment of convective energy (e.g Prein et al., 2013; Hohenegger et al., 2020)"

Prein, A.F., Gobiet, A., Suklitsch, M. *et al*. Added value of convection permitting seasonal simulations. *Clim Dyn* **41,** 2655–2677 (2013). https://doi.org/10.1007/s00382-013-1744-6

Hohenegger, C., L. Kornblueh, D. Klocke, T. Becker, G. Cioni, J. F. Engels, U. Schulzweida, and B. Stevens, 2020: Climate statistics in gobal simulations of the atmosphere, from 80 to 2.5 km grid spacing. J. Meteor. Soc. Japan, 98, 73-91

**16. Ln-547: The statement is valid if the North Atlantic SSTs are strong enough to permit the convection in summer. But in winter, the scenario mostly depends on the Arctic cold air outbreaks and convergence of the moisture from the adjacent areas into the AR.**
Yes. We made this more clear by adding the sentence:
"This is in particular the case during the warm season when SST are high enough to allow convection or during winter when moisture convergence advects moisture from adjacent areas."

**17. Fig-12: Does the inter-model standard deviation means the average standard deviation obtained from different RCA ensembles?**
No. It is the standard deviation across all 9 models (RCA-HAD, RCA-ECE…). We made this more clear in the figure caption by adding "...across all 9 models (CAN, CNRM,...NORESM)"

**18. Ln-635: Please check the sentence as it is contradicting.**
Done. See our response to review report 1 on the same issue.

---

## Author Response (AR3)

**Dear Authors,**

**I am glad to accept your study for publication in Earth System Dynamics subject to a final minor edit. As Reviewer #2 points out in his comment No. 8 (Line 225), there are some caveats to the chosen percentile threshold methodology. While I agree that changing this methodology would require additional tests beyond the scope of the current submission, I would nonetheless suggest that you briefly mention its limitations as well as potential alternative methodologies (such as the one suggested by the Reviewer) that may be tested in future studies.**

We thank the Editor for careful reading of our manuscript.

We find this is an excellent idea and have included

"We note that the detection using the 85th percentile at 10°W is well validated for AR over the Atlantic. However, it may lead to some inaccuracies over the eastern European land mass. This is because the AR landfall at the western European boundary may decrease the IVT along the eastern landmass due to moisture loss by precipitation and the moisture cut-off from the ocean. This could limit the detection of AR impact over the distant parts in eastern Europe. A potential solution could be to take the local 85th percentile over land points instead the 85th percentile at 10°W as threshold. However, this should be robustly tested and validated in future research."

At line 230 of the revised manuscript.

Kind regards
Matthias